# A comparison of vertebral preparation techniques for increasing reader precision and agreement in vertebral band pair identification of Northeast Atlantic skates

Eleanor S. I. Greenway[1,2]*, Lorenzo L. Elias[3], Antonella Consiglio[4], Andrea Bellodi[5,6], Blondine Agus[5,6], Jurgen Batsleer[2], Karen Bekaert[7], Pierluigi Carbonara[4], Manfredi Madia[6,8], Mauro Sinopoli[8], Michele Palmisano[4], Ilse Maertens[7], Jan Jaap Poos[1,2]

**1** Aquaculture and Fisheries Group, Wageningen University and Research, Wageningen, The Netherlands, **2** Wageningen Marine Research, Haringkade 1, 1976 CP, IJmuiden, The Netherland, **3** Marine and Estuarine Biodiversity, Arcadis, Piet Mondriaanlaan 26, 3812 GV, Amersfoort, The Netherlands, **4** COISPA Tecnologia & Ricerca, Stazione Sperimentale per lo Studio delle Risorse del Mare, Bari, Italy, **5** Department of Integrated Marine Ecology, Calabria Marine Center, Stazione Zoologica Anton Dohrn Napoli (SZN), C.da Torre Spaccata,Amendolara, Italy, **6** Department of Life and Environmental Sciences, University of Cagliari, via T. Fiorelli 1 Cagliari, Italy, **7** Flanders Research Institute for Agriculture, Fisheries and Food (ILVO), Animal Sciences Unit, Fisheries and Aquatic Production, Jacobsenstraat 1, Ostend, Belgium, **8** Sicily Marine Centre, Stazione Zoologica Anton Dohrn, Lungomare Cristoforo Colombo, Palermo, Italy

* eleanor.greenway@wur.nl

## Abstract

Direct ageing of elasmobranch species comes with many challenges where the success of preparation methods tends to be species-specific. In this study, we compare age estimations from different preparation methods concerning vertebral location (anterior and posterior), vertebral structure (whole and sectioned), and vertebral staining (stained and unstained) for three skate species: *Raja brachyura*, *Raja clavata*, and *Raja montagui*. Age estimations were derived from modal ages of eight age readers from Dutch, Belgian, and Italian institutions. Only vertebrae from the same individual, where both conditions of the preparation method were available, were used in the analysis. Precision measures based on modal ages were variable but consistent with other elasmobranch ageing studies. For all species, anterior vertebrae showed lower CV values on average compared to posterior vertebrae. APE and CV were lower whole vertebrae compared to sectioned vertebrae regardless of vertebral location or staining, for all species. Age bias plots showed age estimations were significantly higher (p < 0.05) in anterior vertebrae compared to posterior vertebrae regardless of vertebral structure or staining for *R. brachyura* and *R. clavata*. Age estimations were similar for whole and sectioned anterior vertebrae for *R. clavata* and *R. montagui*, and the effect of staining was variable but minimal across all species. Principal component analyses showed the vertebral preparation method had

**Data availability statement:** All data files are available from the DANS Data Station Life Sciences database https://doi.org/10.17026/LS/ARNWEH.

**Funding:** This work was supported by the 'Bridging knowledge gaps for sharks and rays in the North Sea' project, funded by the European Maritime, Fisheries and Aquaculture Fund https://oceans-and-fisheries.ec.europa.eu/funding/emfaf_en. The funders had no role in study design, data collection and analysis, decision to publish, or preparation of the manuscript.

**Competing interests:** The authors have declared that no competing interests exist.

little effect on the variability of the data. The results from this study show the use of anterior, whole, unstained vertebrae were more precise for age classes of 0–9 years, offering significant reductions in both preparation time and cost by eliminating the need for sectioning or staining. With this study we aim to provide some consistency among future ageing studies concerning *Raja* species which, in turn, improves data quality and management.

## 1. Introduction

Estimating life-history characteristics such as mortality in populations and individual growth in aquatic populations relies on accurate age determination of individuals, which is notoriously challenging [1,2]. As a result, ageing is the subject of ongoing research where several techniques have been developed to improve the accuracy and reliability of age estimations [3,4]. Otoliths are widely recognised as the most accurate structure for age estimation in teleost fish [5], but these structures are absent in cartilaginous fish such as sharks, skates, and rays (elasmobranchs). Alternative calcified structures, such as spines, rayfins, caudal thorns, and vertebrae, often feature patterns that reflect annual growth bands and are therefore used for age estimation in these species [6,7]. In teleost fish, age estimations from otoliths and vertebrae have been compared, demonstrating that vertebrae produce ages comparable to those of otoliths [8–10].

Despite the fact that elasmobranchs remain a largely understudied group, there is an abundance of different ageing techniques [4,11] which have been used across multiple elasmobranch species. Currently, vertebrae used for ageing in elasmobranchs are either examined as whole structures or are sectioned. Staining is then often used to optimise the visibility of growth increments [12], where a range of staining methods have been tested [13] including alizarin red [4,14,15], silver nitrate [16,17], cedar oil [4,18], and crystal violet [17,19–21]. Research over the past decade continues to refine or adapt these staining techniques [22–24]. However, staining is not always applied [25–27] and can sometimes be ineffective [6,28], where recent studies make use of editing software to digitally enhance growth bands [27].

Although age estimation in elasmobranchs has been validated in several studies [29–31], vertebral ageing cannot be universally confirmed because band visibility and periodicity vary among species; this discrepancy reflects interspecific differences in the biogeochemistry of vertebral cartilage, due to variation in matrix composition, degree of mineralization, and trace-element incorporation—factors influenced by diet and environmental conditions (e.g., temperature, salinity, dissolved oxygen, pH/carbonate chemistry, depth) [32,33]. Moreover, the use of vertebrae for age estimation has been invalidated for several species, including the common sawshark (*Pristiophorus cirratus*) [34] and the Pacific angel shark (*Squatina californica*) [35]. As a result, a single standardized technique for age estimation across all elasmobranchs remains unlikely [6,36].

Comparisons of vertebral preparation methods have shown variable results among elasmobranch species. Previous research shows that sectioned vertebrae produce higher and more accurate ages compared to whole vertebrae in as seen in the spot-tail shark (*Carcharhinus sorrah*), the Australian blacktip shark (*Carcharhinus tilstoni*) [36], and the Port Jackson shark (*Heterodontus portusjacksoni*) [37]. In contrast, other studies show that whole vertebrae produce similar ages to sectioned vertebrae in young blue shark (*Prionace glauca*) [38], or may even be preferred over sectioning vertebrae as in the crocodile shark (*Pseudocarcharias kamoharai*) [39]. Regardless, age underestimations occur in both whole and sectioned vertebrae in older individuals as growth bands either become tightly grouped on the vertebral edge making them more difficult to distinguish [40] or growth bands deposition halts once somatic growth slows or ceases [24,41]. Such age underestimations have been validated from tagging studies [36,42,43], or bomb radiocarbon dating [44,45]. Collectively, these studies highlight both the uncertainty and limitations of using vertebral centra for age estimations and demonstrate that preparation methods are highly species-specific. However, vertebral ageing appear most reliable for shorter-lived species (where maximum estimated age is less than 20 years) for which age estimates have been validated [36,40,42,46].

Studies regularly use larger anterior, thoracic vertebrae [47] in ageing because their size facilitates age reading [4,11]. One study used six individuals to compare age estimations from anterior and posterior vertebrae of Pacific spiny dogfish (*Squalus suckleyi*), where no significant difference in the estimated ages was found [48]. However, other studies on different shark (*Squatina dumeril, Carcharodon carcharias, Lamna nasus, Isurus oxyrinchus, Alopias vulpinus, Prionace glauca, and Carcharhinus obscurus*) [41], skate, (*Leucoraja erinacea, Leucoraja ocellata, Dipturus laevis*), and ray species (*Dasyatis sabina*, and *Urobatis halleri*) [24] have reported age underestimations in posterior vertebrae as centrum morphology and band-pair counts vary along the vertebral column [24,41]. Nonetheless, sampling posterior vertebrae has practical advantages as individuals obtained from market sampling events still retain commercial value and can be sold, thereby reducing both food waste and sampling costs.

In the Northeast Atlantic, a number of *Raja* species are commercially fished, where age data are fundamental for fisheries management. Currently, the scientific advice for the total allowable catches (TAC) for these populations are based on time series of catch rates in annual research vessel surveys [49,50], or from surplus production models combining these catch rates with total fisheries catches [50,51]. The TACs set by the EU are for groups of skates and rays, by combining the single species advice into a grouped TAC [52]. The scientific advice and final decisions on the catches of the species thus disregard a number of key life-history characteristics of the populations. Advancements in age estimation, particularly through refined methods for examining vertebrae, offer better insights into species' growth rates, longevity, and reproductive patterns.

Several studies have identified growth curves for *Raja* species in the Northeast Atlantic using both whole [53–57] and sectioned [20,29,56,58,59] vertebrae. These studies indicate maximum ages of less than 20 years, and are supported by long term tagging studies [60]. However, a comprehensive comparison of different vertebral preparation methods does not exist for these species. Comparisons of growth estimates among these studies is therefore hampered by the potential differences in estimated ages resulting from the preparation method used. Crystal violet has been identified as the most successful staining technique for *Raja* species [61], and has yielded successful results for *Raja* species in more recent studies [55,56]. The aim of this study was to compare age estimates derived from anterior and posterior vertebrae, whole and sectioned vertebrae, and stained and unstained vertebrae of *Raja brachyura*, *Raja clavata*, and *Raja montagui*. By comparing several species, we aim to draw conclusions on the most effective vertebral preparation techniques for the *Raja* genus.

## 2. Methods

### 2.1. Sampling

Individuals of *R. brachyura*, *R. clavata*, and *R. montagui* were collected between June 2018 and March 2024. Samples and data were collected either from landed individuals intended for human consumption, or as part of routine data

collection in a commercial fisheries program for discard sampling in The Netherlands. This study was non-experimental as samples obtained from discard sampling involved opportunistically collecting individuals that were already deceased, where The Dutch Experiments on Animals Act does not apply. An ethical review by the Statement Animal Experiment Committee was therefore not required. No additional mortality or animal discomfort beyond standard fishing operations was caused for sample collection for the purpose of this study.

Total length ($L_T$) (cm), disc width (cm), weight (g), and sex were recorded. Locations from discard sampling events were used directly, while locations of samples obtained from fish auctions were estimated by calculating the centroids of ICES areas based on fishing activity reported in mandatory fisheries catch and effort logbooks, as stored at Wageningen University and Research, and the Centre for Fisheries Research (Fig 1). Specimens were stored at –20°C prior to dissection and maturity stage was determined from macroscopic examination of the reproductive organs [62]. Vertebral samples were collected during dissection and subsequently frozen at until further analysis.

## 2.2. Vertebrae preparation

### 2.2.1. Vertebral location.
In elasmobranchs, the occipital hemicentrum is the first vertebra of the spine lacking a centrum [63]. The first counted anterior vertebra was the first fully developed vertebra after the occipital hemicentrum. The total number of vertebrae identified in this study differed for the three species: 139, 131, and 127, for *R. brachyura*,

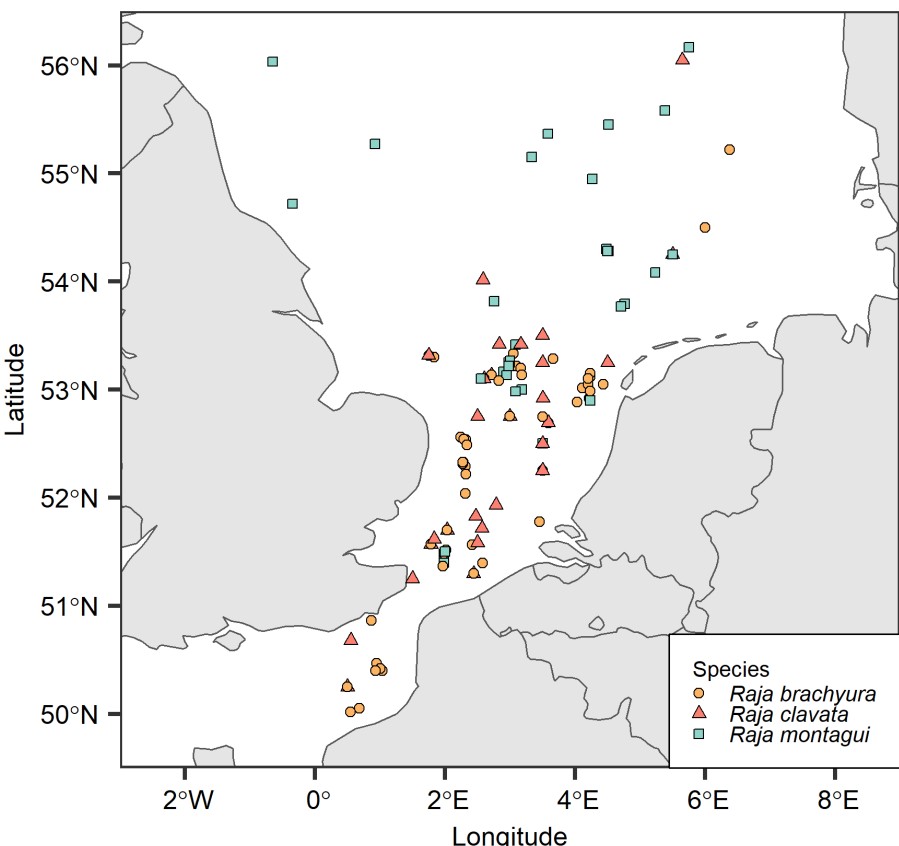

**Fig 1. Capture locations of *Raja brachyura*, *Raja clavata*, and *Raja montagui*.** Samples were collected between June 2018 and March 2024. Locations were estimated from the national fisheries catch and effort database by linking fish landings with fishing trip locations and visualised using the "*rnaturalearth*" package in R version 4.4.1.

*R. clavata*, and *R. montagui*, respectively. The development of the haemal arches in the tail also occurred at different placements in the spine; occurring between the 31st–35th, 24th–28th, and 26th–29th vertebrae for *R. brachyura, R. clavata,* and *R. montagui*, respectively. Anterior vertebrae were sampled between the 11th and 15th vertebrae for all species. However, due to the differences in morphology, samples from posterior vertebrae were sampled between the 52nd–58th, 49th–55th, and 47th–53rd vertebrae for *R. brachyura, R. clavata,* and *R. montagui*, respectively.

**2.2.2. Vertebral cleaning.** A minimum of five vertebra were excised from anterior and posterior sections which were immersed in hot (~80°C) water to remove excess cartilage or connective tissue. Remaining intervertebral tissue was removed by submersion in a pepsin solution (1.46% +/- 2%), prepared by adding 8 ml of hydrochloric acid (25%) to one liter of tap water preheated to 46–48°C, followed by 15 ml pepsin (660–800 European Pharmacopoeia units/ml) [55]. Vertebrae were immersed in the pepsin solution for up to 30 minutes, depending on the vertebral size, whilst continuously stirring on a heat plate at a constant temperature (44–46°C). Samples were then rinsed in hot water (~80°C) for 5 minutes to stop the enzyme reaction. Vertebrae were bathed in 70% ethanol for 10 minutes, air dried, then etched in ethylenediaminetetraacetic acid (EDTA, 5%) for 10 minutes and subsequently rinsed in running water for 10 minutes [20]. After cleaning, vertebrae from the anterior and posterior sections either remained as whole or were subsampled and sectioned prior to staining.

**2.2.3. Vertebral structure and staining.** Vertebrae were sectioned using previously described methods [6], where vertebrae were embedded in black epoxy and sagittally sectioned with the use of an ATM Brilliant 250.3 diamond saw. Sections were created with a thickness between 550 μm and 600 μm. Whole and sectioned vertebrae were either left unstained or underwent staining (Fig 2). A solution of crystal violet (0.005%, with distilled water) was used to stain vertebrae for a maximum of two hours, rinsed with running water for two minutes, then air dried. 70% ethanol was used to remove stains in overstained vertebrae, if necessary.

## 2.3. Age readings

For each individual, one vertebra was photographed for each of the preparation methods, using a stereomicroscope (LAS; Leica Microsystems, Switzerland) captured at a minimum resolution of 0.8 MP (1042 X 768). Vertebrae were immersed in water and photos were taken in greyscale to enhance images [6]. Images were uploaded to ICES SmartDots software (v4.1) where ages were determined by counting pairs of opaque and translucent bands, starting with the first band pair after the birthmark. Birthmarks were easily identified in both whole and sectioned vertebrae as the first distinct opaque band distal to the focus [64] (Fig 2). In sectioned vertebrae, this is sometimes associated with an angle change in the *corpus calcareum*, although it is not always evident [6]. Age estimations were based on band counts, which have been validated for *R. clavata* [29]. Each vertebra was assigned an age with yearly resolution, using January 1 as the birthday, and given an AQ score ranging from AQ1 (highest clarity) to AQ3 (unreadable) indicating the quality of age reading. Age estimates were carried out independently by eight readers representing institutes in The Netherlands, Belgium, and Italy. All age readers attended the ICES workshop on age reading and maturity stages for elasmobranch species (WKARMSE), where age readers were familiarised with age reading protocols for both whole and sectioned vertebrae. Modal ages measured in years (yr) for each skate were obtained considering age estimations from all readers for each preparation method. When 2 or more modal ages were present, the age was assigned according to the age estimated by the most experienced reader. Any vertebrae for which no modal age could be derived were removed from further analyses. The use of modal age can potentially introduce a systematic underageing bias, particularly if the highest (oldest) age estimates more closely approximate true age. However, the primary objective of this study was not to determine absolute accuracy, but rather to assess how different preparation methods influence the readability and consistency of vertebral age estimates across multiple readers. The modal age approach enables quantification of reader consensus and exclude indeterminate cases (i.e., instances with no clear mode), thereby focusing the analysis on relative precision rather than absolute accuracy.

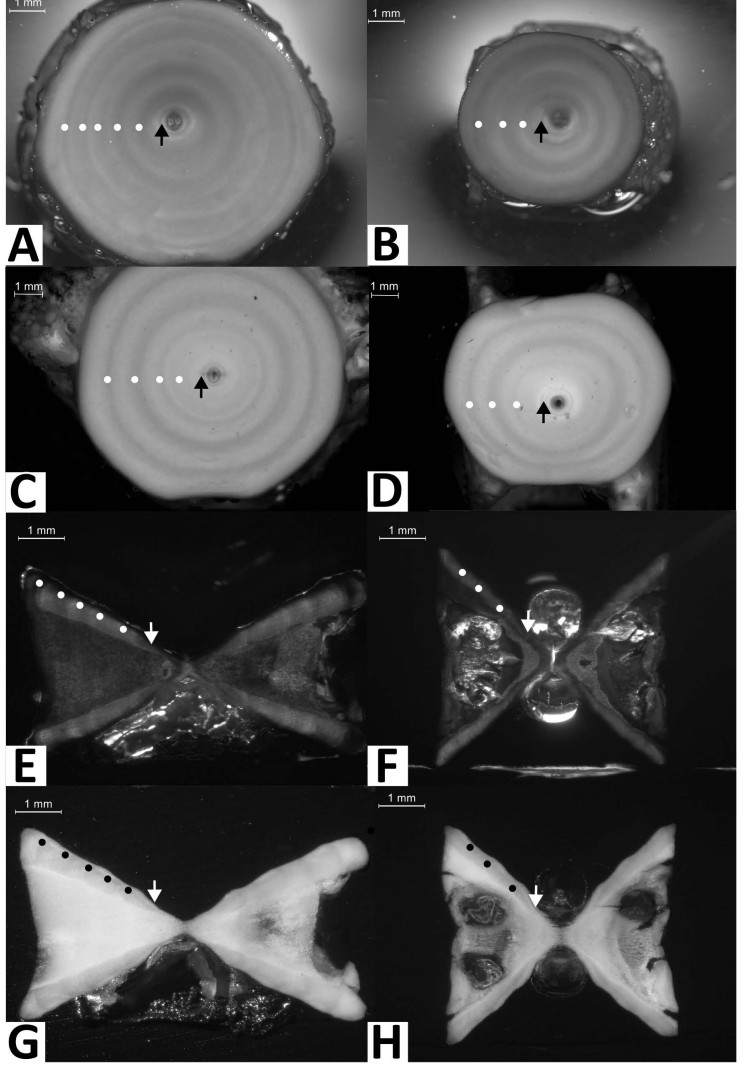

**Fig 2. Example images of vertebral centra for each preparation method.** Anterior whole stained **(A)**, posterior whole stained **(B)**, anterior whole unstained **(C)**, posterior whole unstained **(D)**, anterior sectioned stained **(E)**, posterior sectioned stained **(F)**, anterior sectioned unstained **(G)**, posterior sectioned unstained (H) from *Raja brachyura*. Images C and D were from a were from an 85.2 cm $L_T$ female caught in January 2022, all other images were from a 103.7 cm $L_T$ male caught in May 2023. Annual growth rings with complete opaque bands were annotated, with the birthmark indicated by an arrow.

### 2.4. Preparation method comparison and ageing precision

Modal ages were compared across all preparation methods: vertebral location (anterior and posterior), vertebral structure (whole and sectioned) and vertebral staining (unstained and stained), using a 2 x 2 x 2 factorial design (Fig 3). Pairwise comparisons were conducted with a Mann–Whitney–Wilcoxon test and presented in age bias plots.

Measures of precision assess the ease or the reproducibility of age estimations in a particular structure [65]. Precision was determined using average percent error (APE), coefficient of variation (CV), and the percentage of agreement (PA). *APEj* is the absolute deviation from the mean age of the *j*th fish

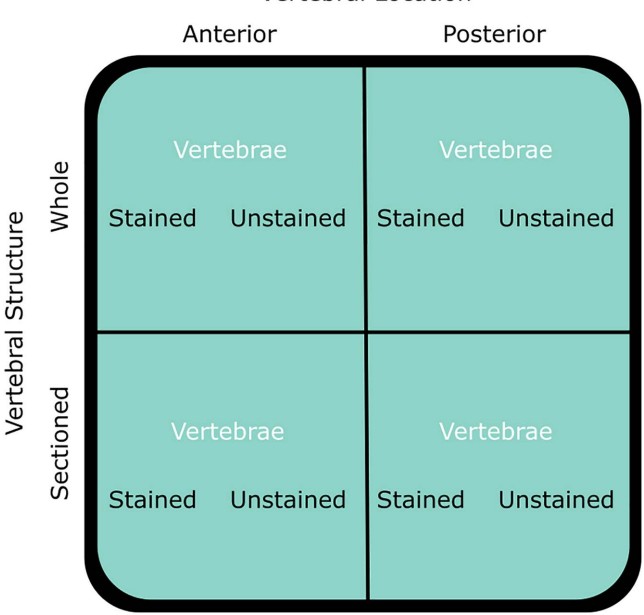

Vertebral Location

**Fig 3. Schematic representing the 2 x 2 x 2 factorial design.** Vertebrae from the same individual were used in one or more of the preparation methods described. Differences in the age estimations for each treatment were tested using a Mann–Whitney–Wilcoxon test. This experimental design was applied consistently across all species: *Raja brachyura*, *Raja clavata,* and *Raja montagui.*

$$APEj(\%) = 100 \ \times \ \frac{1}{R}\sum\nolimits_{i=1}^{R} \frac{[Xij - Xj]}{Xj}, \tag{1}$$

where $R$ is the number of reads, $Xij$ is the $i$th age determination of the $j$th fish and $Xj$ is the mean age calculated for the $j$th fish [66]. The coefficient of variation was estimated by

$$CVj\ (\%) = 100 \ \times \ \sqrt{\sum\nolimits_{i=1}^{R} \frac{\frac{(Xij - Xj)^2}{R-1}}{Xj}}, \tag{2}$$

where $CVj$ is the ratio of the standard deviation to the mean age of the $j$th fish, $R$ is the number of reads, $Xij$ is the age determination of the fish ($j$) by reader ($i$) and $Xj$ is the mean age of the $j$th fish [67]. Agreement was calculated as

$$PA = \frac{\sum |\, n_{diff} \leq 1\,|}{n}, \tag{3}$$

where $n_{diff}$ is the difference in age estimations with modal age. Precision of ages and reader agreement were estimated for all eight readers, across all preparation techniques.

## 2.5. Growth model comparison

In order to determine whether vertebral location (anterior and posterior vertebrae) yielded different growth curves, age–at–length data gathered from specimens for which ages were given for both preparation method were modelled and the resulting curves compared. In growth model calculations, whole stained vertebrae were used for consistent methodology, and sexes were grouped due to the low number of large *R. clavata* and *R. montagui* males. Growth models were conducted

exclusively to test whether the two types of vertebrae provided the same results in terms of growth curves. For this reason, we decided to rely only on the von Bertalanffy function, as this is more easily comparable with the existing literature.

Curve fitting and growth model reliability near age 0, age–at–length data from eight *R. clavata* and ten *R. montagui* hatchlings born in captivity in the aquatic research facility of Wageningen University CARUS were added to the dataset. These observations were included used to constrain the early proportion of the growth curve, with the parameter *t0* treated as a mathematical curve-fitting parameter. The von Bertalanffy nonlinear growth model [68] was applied to the age-at-length data with the following equation

$$L_{Tt} = L_{T\infty} \left(1 - e^{-k(t-t^0)}\right),$$

(4)

where *t* is the estimated age, $t^0$ is the hypothetical age of an individual with $L_T$ equal to 0, $L_{T\infty}$ is the species' theoretical maximum length, and k is the growth coefficient. Growth patterns were modelled through the FSA package (ver. 0.9.5) [69] in R environment (ver. 4.4.1) [70]. The Chen test [71] was employed to investigate potential differences between growth curves produced by anterior and posterior vertebrae.

## 2.6. Multivariate analysis

Principal component analysis (PCA) was used to identify the key variables contributing to the variations in geographical location (longitude and latitude), and therefore age determination of the vertebrae for each species. PCAs were performed using the *FactoMineR* library [72] in R [70]. Principle components (PCs) contributing to a cumulative explained variance greater than 60% were retained.

The analysis considered total length (cm $L_T$), age, and geographical location (longitude and latitude) as continuous variables, while species, sex, reader experience, and preparation methods were considered as categorical variables. Age readers were categorised into experience levels based on the total number of vertebral readings where 'low' was less than 2000 vertebral readings, 'medium' was between 2000–5000 vertebral readings and 'high' was more than 5000 vertebral readings. Of the age readers in this study, one had 'low' experience, and one had 'high' experience, all others were 'medium' experienced. The preparation method was composed of three parts: vertebral location (anterior and posterior), vertebral structure (whole and sectioned), and vertebral staining (stained and unstained), resulting in eight different combinations of these factors. Vertebral location, vertebral structure, and vertebral staining were also considered separately in the analyses in order to determine which method explained the most variance in the dataset. The initial analysis was carried out on the whole data set considering all species, then data were subset for analyses of each vertebral preparation method (location, structure, staining), including only individuals for which data were available for both conditions within each preparation method. Only statistically significant correlations were considered, including the Pearson correlation coefficient (R > 0.2), the proportion of the variance in PC explained by a variable ($R^2$), and the relative p–values < 0.05.

## 3. Results

A total of 456 individuals were sampled where 25 individuals were excluded from modal age calculations. Samples had broad length ranges, comprising *R. brachyura* (n = 123; 14–103.7 cm $L_T$), *R. clavata* (n = 221; 20–90 cm $L_T$), and *R. montagui* (n = 87; 11.9–62.5 cm $L_T$). Each of the eight vertebral preparation methods (Table 1) were successfully applied to the vertebrae and read independently by eight age readers.

### 3.1. Ageing precision

Ageing precision showed high variability in age readings for all species and methods. *R. brachyura* (APE 28–55%; CV 25–37%) and *R. clavata* (APE 30–47%; CV 20–36%) show similar values, while values were higher for *R. montagui* (APE

**Table 1. Total number of samples generated for each vertebral preparation method.**

| Species | Anterior sectioned stained | Anterior whole stained | Posterior sectioned stained | Posterior whole stained | Anterior sectioned unstained | Anterior whole unstained | Posterior sectioned unstained | Posterior whole unstained |
|---|---|---|---|---|---|---|---|---|
| *Raja brachyura* | 49 | 129 | 48 | 129 | 48 | 21 | 46 | 21 |
| *Raja clavata* | 32 | 230 | 31 | 211 | 32 | 15 | 31 | 11 |
| *Raja montagui* | 29 | 87 | 22 | 66 | 30 | 14 | 23 | – |

36–65%; CV 32–92%). PA had some variability among species which ranged from 36–55%, 42–59%, and 46–61%, for *R. brachyura, R. clavata*, and *R. montagui* respectively (Table 2).

Both APE was lower by 2–19% and CV by 3–25% for anterior, whole, stained vertebrae compared to sectioned vertebrae for all species. PA was also consistently higher by 13–14% for whole compared to sectioned vertebrae. In most cases, anterior vertebrae resulted in CV values 1–17% lover compared to posterior vertebrae, regardless of vertebral structure or staining (Table 2).

Posterior sectioned vertebrae stained and unstained were the least successful preparation methods with the highest percentage of unreadable (AQ3) age reading attempts 8–10%, 19–25%, and 11–27%) for *R. brachyura, R. clavata*, and *R. montagui* respectively.

### 3.2. Age bias plots

Age bias plots comparing differences in modal ages for vertebral location and vertebral structure (Fig 4) showed age estimations were significantly higher in anterior compared to posterior vertebrae regardless of vertebral structure for *R. brachyura* (sectioned: W = 951, p = 0.014; whole: W = 8008, p = 0.01) and *R. clavata* (sectioned: W = 459, p = 0.004; whole: W = 22513.5, p = 0.009). This pattern was not observed for *R. montagui*, which had a maximum estimated age of 6 (Fig 4A–F).

**Table 2. Summary of ageing precision results of each vertebral preparation method.**

| Species | Precision Measure | Anterior sectioned stained | Anterior whole stained | Posterior sectioned stained | Posterior whole stained | Anterior sectioned unstained | Anterior whole unstained | Posterior sectioned unstained | Posterior whole unstained |
|---|---|---|---|---|---|---|---|---|---|
| *Raja brachyura* | APE (%) | 55 | 48 | 41 | 42 | 41 | 44 | 37 | 28 |
| | CV (%) | 35 | 29 | 37 | 35 | 31 | 25 | 29 | 25 |
| | PA (%) | 36 | 50 | 46 | 49 | 48 | 47 | 53 | 55 |
| | Age range (yr) | 2–8 | 0–7 | 1–6 | 0–7 | 1–8 | 2–6 | 2–5 | 1–7 |
| *Raja clavata* | APE (%) | 42 | 40 | 34 | 40 | 36 | 47 | 30 | 30 |
| | CV (%) | 32 | 29 | 36 | 30 | 27 | 23 | 31 | 20 |
| | PA (%) | 42 | 55 | 44 | 59 | 49 | 49 | 47 | 53 |
| | Age range (yr) | 2–6 | 0–9 | 1–5 | 0–6 | 2–7 | 1–6 | 1–5 | 1–5 |
| *Raja montagui* | APE (%) | 55 | 36 | 56 | 35 | 60 | 37 | 65 | – |
| | CV (%) | 57 | 32 | 56 | 32 | 77 | 36 | 92 | – |
| | PA (%) | 46 | 59 | 56 | 61 | 57 | 50 | 53 | – |
| | Age range (yr) | 0–5 | 0–6 | 0–4 | 0–4 | 0–6 | 0–5 | 0–4 | – |

The age range (yr) in the precision measures column reflects the age range of the modal ages as estimated by all age readers.

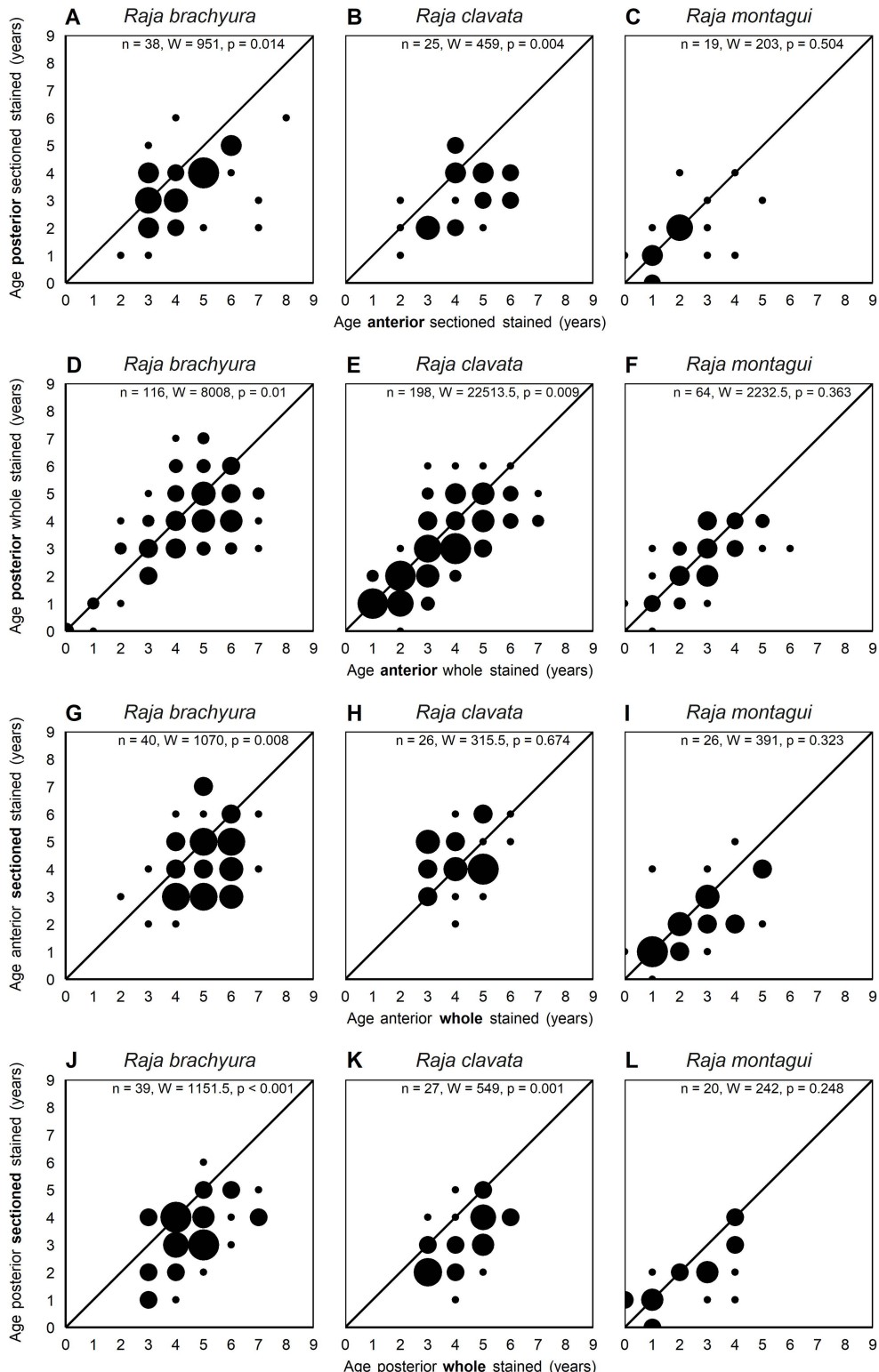

**Fig 4. Age bias plots comparing modal age estimations (years) between different vertebral preparation methods.** Anterior, sectioned, stained vertebrae and anterior, whole, stained vertebrae **(A–C)**; posterior, sectioned, stained vertebrae and posterior, whole, stained vertebrae **(D–F)**; anterior, whole, stained vertebrae and posterior, whole, stained vertebrae **(G–I)**; anterior, sectioned, stained vertebrae and posterior, sectioned, stained vertebrae **(J–L)**, for *Raja brachyura*, *Raja clavata* and *Raja montagui*. Significance was tested using a Mann–Whitney–Wilcoxon test where W is the test statistic.

No significant differences were observed between the vertebral structure of anterior vertebrae for *R. clavata* and *R. montagui*. However, modal ages were significantly higher in whole vertebrae compared to sectioned vertebrae for *R. brachyura* (W = 1070, p = 0.008) (Fig 4G–I). When comparing modal ages of the vertebral structure of posterior vertebrae, whole vertebrae also resulted in significantly higher age estimates for *R. brachyura* (W = 1151.5, p < 0.001) and *R. clavata* (W = 549, p = 0.001), but not for *R. montagui* (Fig 4J–L).

Age bias plots comparing the vertebral staining showed mixed results (Fig 5). In most cases, there were no significant differences in age estimation between stained and unstained vertebrae for either location or structure. Significant differences in modal ages were only observed for anterior sectioned vertebrae for *R. montagui* (W = 464, p = 0.017), anterior whole vertebrae for *R. brachyura* (W = 247.5, p = 0.042), and posterior whole vertebrae for *R. brachyura* (W = 225.5, p = 0.038), in these instances ages were significantly higher in stained vertebrae (Fig 5A, C, D, J).

### 3.3. Growth modelling

Whole stained vertebrae were used to model differences in growth patterns between anterior and posterior vertebrae. A total of 396 specimens were analysed, specifically 116 *R. brachyura* (17.5–103.7 cm $L_T$), 206 *R. clavata* (12.8–90.0 cm $L_T$), and 72 *R. montagui* (11.2–62.5 cm $L_T$). The maximum age estimated from posterior vertebrae was lower than age estimations from anterior vertebrae both for *R. clavata* (6 against 7 years) and *R. montagui* (4 against 6 years) (Table 3). Conversely, the same maximum age (7 years) was estimated for *R. brachyura* from both vertebral locations (Table 3).

Anterior vertebrae produced growth curves characterized by a higher asymptotic length ($L_{T\infty}$) and lower growth coefficients(k) for *R. clavata* and *R. montagui*, while growth curves for *R. brachyura* resulted in very similar von Bertalanffy's parameters (Table 3; Fig 6). However, the Chen test found these differences to be significant only in *R. clavata* (Fobs. > Fcrit; p < 0.001) (Table 3).

### 3.4. Multivariate analysis

The PCA was based on a correlation matrix of quantitative and supplementary qualitative variables, where the first two PCs were retained, explaining more than 76% of the total variance in all cases. The major contributions to the first component (PC1; 50.9%) were given by total length and Latitude. Age and longitude gave the highest contribution to the second component (PC2; 35.1%) (Table 4). Species showed the highest correlation with PC1 where differences were observed between the three species (Fig 7). PCAs were calculated after selecting for individuals where both conditions were available for each preparation method; vertebral location (Table 5; Fig 8), vertebral structure (Table 6; Fig 9), and vertebral staining (Table 7; Fig 10).

**3.4.1. Vertebral location.** Whole, sectioned, and stained vertebrae were selected where both anterior and posterior vertebrae were analysed from the same individual. The total explained variance was 85%, 89.4%, and 86.6% for *R. brachyura, R. clavata,* and *R. montagui* respectively, where longitude and latitude were the major contributors to PC1 for *R. brachyura* and *R. clavata* (Table 5; Fig 8). Age and total length had strong negative correlations with PC1 for *R. brachyura* and *R. clavata*, but positive correlations for *R. montagui*. All supplementary qualitative variables, including vertebral structure and vertebral location had no significant effect or explained little of the total variance in either PCs, for all species (Table 5; Fig 8).

**3.4.2. Vertebral structure.** Anterior, stained vertebrae were selected where both whole and sectioned vertebrae were analysed from the same individual. The total explained variance was 78.9%, 73.8%, and 87.8% for *R. brachyura, R. clavata,* and *R. montagui* respectively, where longitude and latitude were the major contributors to PC1 and age was the major contributor to PC2 for *R. brachyura* and *R. clavata* (Table 6; Fig 9). Age and total length showed negative correlations with PC1 for *R. brachyura* and *R. clavata*, but had the opposite effect for *R. montagui*. All supplementary qualitative variables, including vertebral structure, had no significant effect or explained little of the total variance in either PCs, for all species (Table 6; Fig 9).

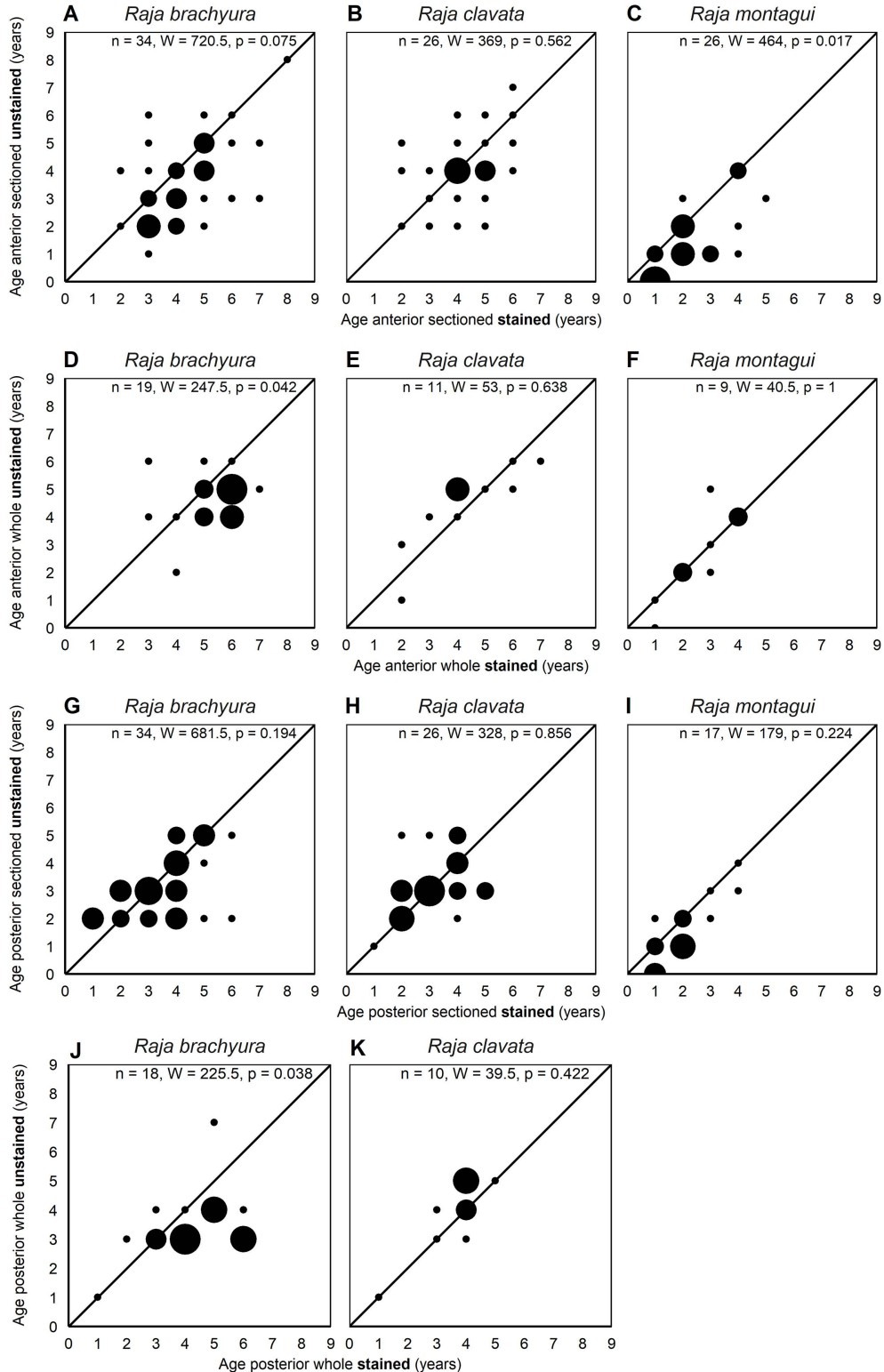

**Fig 5. Age bias plots comparing modal age estimations (years) between stained and unstained vertebral preparation methods.** Anterior, whole, stained vertebrae and anterior, whole, unstained vertebrae **(A–C)**; anterior, sectioned, stained vertebrae and anterior, sectioned, unstained vertebrae **(D–F)**; posterior, sectioned, stained vertebrae and caudal, sectioned, unstained vertebrae **(G–I)**; posterior, whole, stained vertebrae and posterior, whole, unstained vertebrae **(J–K)**, for *Raja brachyura*, *Raja clavata* and *Raja montagui*. Significance was tested using a Mann–Whitney–Wilcoxon test, where W is the test statistic.

**Table 3. Growth parameters (mean ± 95% confidence interval) calculated for *Raja brachyura*, *Raja clavata*, and *Raja montagui*.**

| Species | $L_T$ range (cm) | Age range (years) | Vertebral location | $L_T \infty$ | k | $t^0$ | Chen test (α = 0.05) | p-value |
|---|---|---|---|---|---|---|---|---|
| *Raja brachyura* | 17.5–103.7 | 0–7 | Anterior | 90.44 ± 9.67 | 0.36 ± 0.13 | −0.59 ± 0.27 | Fobs < Fcrit | 0.35 |
| | | 0–7 | Posterior | 92.04 ± 9.54 | 0.37 ± 0.12 | −0.67 ± 0.22 | 1.11 < 2.64 | |
| *Raja clavata* | 12.8–90.0 | 0–7 | Anterior | 101.16 ± 18.80 | 0.21 ± 0.07 | −0.84 ± 0.37 | Fobs > Fcrit | 0.0004* |
| | | 0–6 | Posterior | 84.58 ± 7.87 | 0.36 ± 0.08 | −0.54 ± 0.24 | 7.84 > 2.63 | |
| *Raja montagui* | 11.2–62.5 | 0–6 | Anterior | 64.90 ± 13.47 | 0.37 ± 0.19 | −0.79 ± 0.83 | Fobs < Fcrit | 0.67 |
| | | 0–4 | Posterior | 58.07 ± 7.83 | 0.58 ± 0.24 | −0.54 ± 0.41 | 0.52 < 2.67 | |

$L_{T\infty}$ is the maximum asymptotic length, k is the growth coefficient, and $t^0$ is the theoretical age at which $L_T$ equals to zero. The Chen test results are also shown along with their statistical significance.

**3.4.3. Vertebral staining.** Anterior, whole vertebrae were selected where both stained and unstained vertebrae were analysed from the same individual. The total explained variance was 82.4%, 89.2%, and 86.1% for *R. brachyura, R. clavata,* and *R. montagui* respectively, where longitude and latitude were the major contributors to PC1 for *R. brachyura* and *R. clavata,* while age and total length were the major contributors to PC1 for *R. montagui* (Table 7; Fig 10). Age and total length showed strong negative correlations with PC1 for *R. clavata*, but age had a non-significant effect for *R. brachyura*. Unlike the other species, longitude and latitude also showed non-significant effects PC2 for *R. brachyura*. All supplementary qualitative variables, including vertebral staining had no significant effect or explained little of the total variance in either PCs, for all species (Table 7; Fig 10). The results from the PCA analyses show that all qualitative variables (sex, reader experience, and preparation method) had little effect on the variability of the data for all species for all preparation method comparisons (Tables 4:7).

## 4. Discussion

In this study, we refined and compared age estimates derived from different preparation methods concerning vertebral location, vertebral structure, and vertebral staining of *R. brachyura*, *R. clavata*, and *R. montagui* to identify the best current guidelines for ageing *Raja* species. Based on our findings, the preparation method involving the use of anterior, whole, unstained vertebrae proved to be the most precise in this study for all species. Nonetheless, the use of modal age may introduce an under-ageing bias, particularly when the oldest estimates better approximate true age. Our aim, however, was to evaluate how preparation methods influence readability and inter-reader consistency rather than absolute accuracy. While this approach emphasises relative precision, it may overlook older estimates that more closely reflect true age, a limitation relevant to long-lived elasmobranchs. Although the species examined are not among the most long-lived, under-ageing remains a recognised source of bias. Therefore, our finding that anterior, whole, unstained vertebrae provided the highest reader precision and agreement should be interpreted within this context.

### 4.1. Vertebral location

Variability of intra-individual age estimations along the vertebral column have been observed for a number of shark [41,48,73], ray [24], and skate species (this study) where age estimations from anterior vertebrae were generally higher compared to posterior vertebrae. Previous studies suggested that differences in age readings were attributed to the lower precision and readability of posterior vertebrae [7,73,74]. However, recent research shows that vertebrae differ in their structure and in size along the vertebral column, resulting in different rates of band deposition [24,41]. Band counts tend to be similar along the vertebral column for younger individuals as the proportional difference in vertebral size is smaller with more uniform somatic growth [24,41]. This may explain the similarities of age estimations observed between anterior and posterior vertebrae in *R. montagui*, which has a smaller maximum attainable length compared to the other two species.

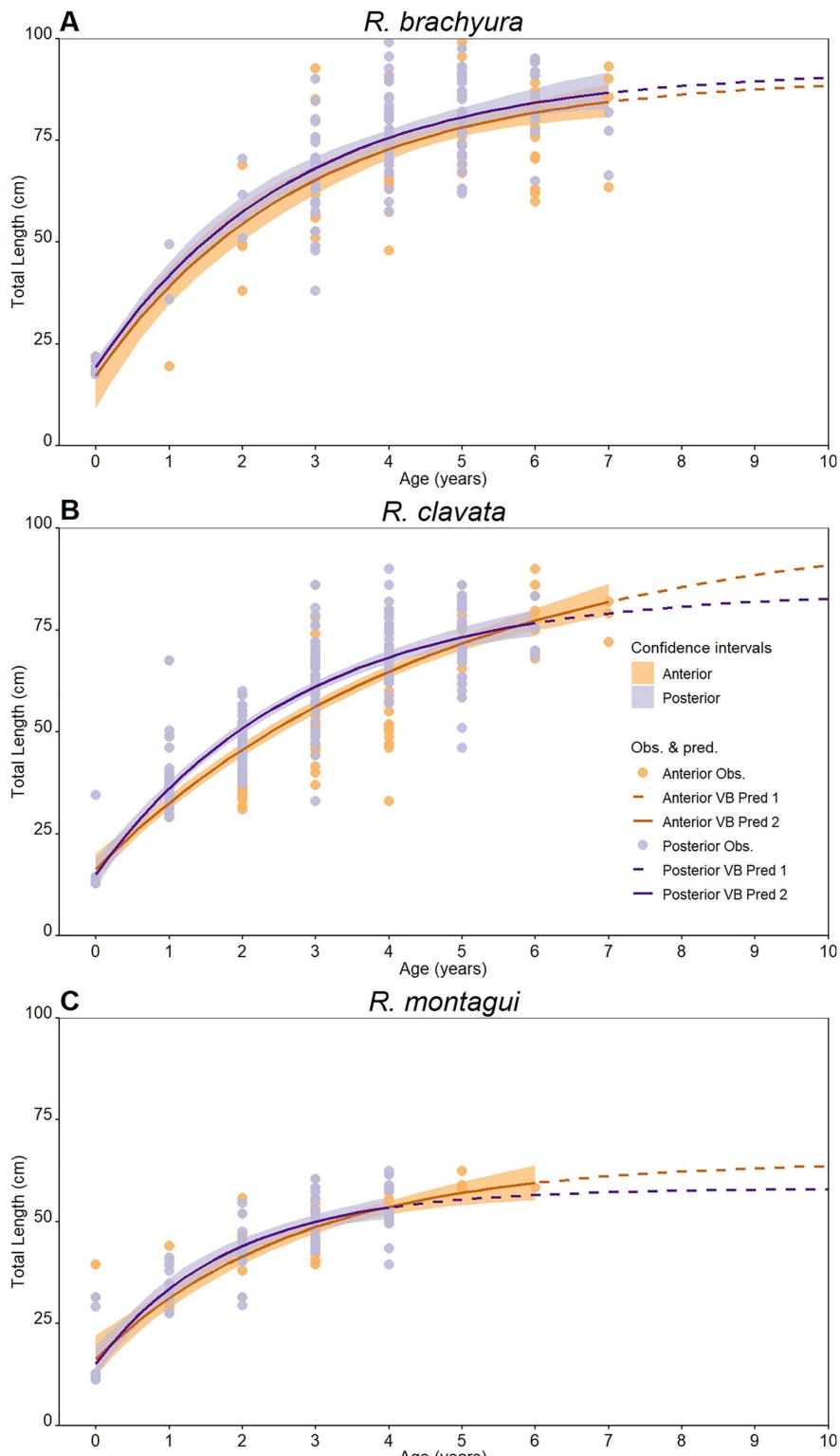

**Fig 6. Von Bertalanffy growth curves.** Constructed using age-at-length data derived from whole, stained vertebrae collected from both anterior and posterior vertebral locations for **(A)** *Raja brachyura,* **(B)** *Raja clavata,* and **(C)** *Raja montagui.*

**Table 4. PCA values of all vertebral preparation methods for the three skate species *Raja brachyura*, *Raja clavata* and *Raja montagui*.**

| Variable | PC1 | PC2 |
|---|---|---|
| | R | R |
| Age | 0.688*** | 0.609*** |
| Total Length | 0.739*** | 0.552*** |
| Longitude | −0.630*** | 0.694*** |
| Latitude | −0.788*** | 0.495*** |
| | R² | R² |
| Species | 0.251*** | 0.073*** |
| Preparation method | 0.035*** | 0.035*** |
| Reader experience | 0.002** | 0.003*** |
| Sex | − | 0.014*** |

Pearson correlation coefficients (R) of the quantitative variables with respect to the first (PC1) and second (PC2) principal components, and the proportion of the variance explained by the supplementary quantitative variables (R²) for each PC. Significance is indicated where p-values are < 0.05*, < 0.01**, or < 0.001***.

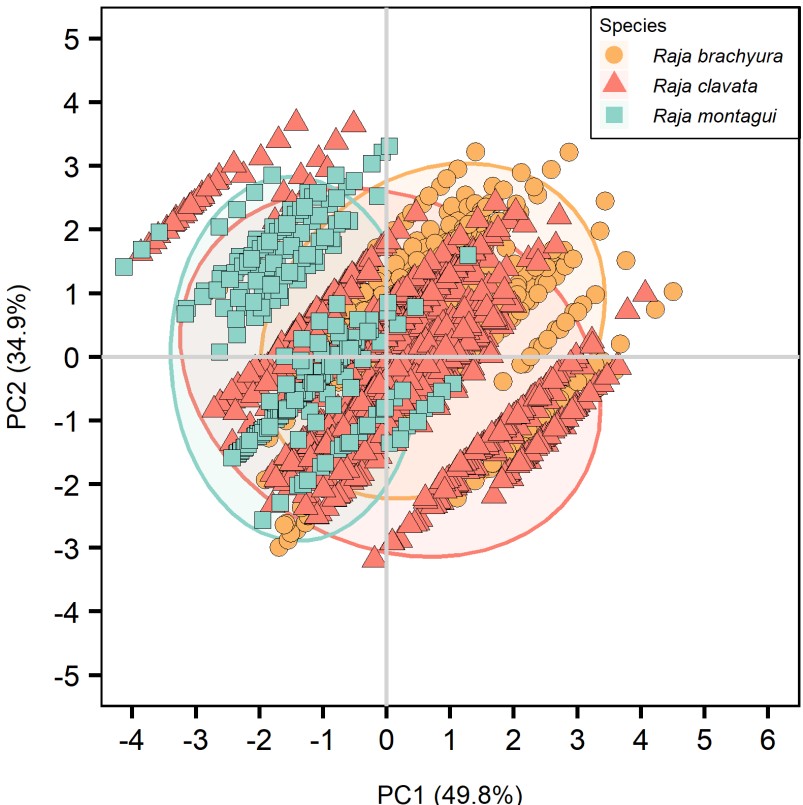

**Fig 7. PCA plot visually representing the three species; *Raja brachyura*, *Raja clavata,* and *Raja montagui*.** Ellipses show 95% confidence intervals.

**Table 5. PCA values calculated concerning vertebral location of whole, sectioned and stained vertebrae for each of the three skate species *Raja brachyura*, *Raja clavata,* and *Raja montagui*.**

| Species | Variable | PC1 | PC2 |
|---|---|---|---|
| *Raja brachyura* | | R | R |
| | Age | −0.507*** | 0.743*** |
| | Total Length | −0.540*** | 0.719*** |
| | Longitude | 0.766*** | 0.553*** |
| | Latitude | 0.854*** | 0.339*** |
| | | R² | R² |
| | Sex | 0.027*** | 0.003*** |
| | Reader experience | – | 0.005* |
| | Preparation method | 0.025*** | 0.006** |
| | Vertebral Structure | 0.023*** | – |
| | Vertebral Location | 0.003* | 0.005*** |
| *Raja clavata* | | R | R |
| | Age | −0.647*** | 0.668*** |
| | Total Length | −0.663*** | 0.652*** |
| | Longitude | 0.734*** | 0.621*** |
| | Latitude | 0.821*** | 0.498*** |
| | | R² | R² |
| | Sex | 0.002* | 0.032*** |
| | Reader experience | – | 0.003** |
| | Preparation method | – | 0.090*** |
| | Vertebral Structure | – | 0.087*** |
| | Vertebral Location | – | 0.003** |
| *Raja montagui* | | R | R |
| | Age | 0.699*** | 0.581*** |
| | Total Length | 0.757*** | 0.524*** |
| | Longitude | 0.844*** | −0.416*** |
| | Latitude | 0.666*** | −0.678*** |
| | | R² | R² |
| | Sex | 0.036*** | 0.032*** |
| | Reader experience | 0.008* | 0.009*** |
| | Preparation method | – | – |
| | Vertebral Structure | – | – |
| | Vertebral Location | – | – |

PCA values of Pearson correlation coefficients (R) of the quantitative variables with respect to the first (PC1) and second (PC2) principal components, and the proportion of the variance explained by the supplementary quantitative variables ($R^2$) for each PC. Significance is indicated where p-values are <0.05*, <0.01**, or < 0.001***.

As fish approach their maximum size, growth rates decrease, leading to reduced band pair development [55,56]. Consequently, growth bands at the outer edge of the vertebrae become tightly grouped and more difficult to distinguish, thereby leading to potential age underestimations [4]. Band pair development has also been shown to be more dependent on body size where the number of growth bands is limited according to the maximum body size rather than continuously accumulating over time [24,41]. *Raja* species are expected to be short-lived species and the lack of band pair

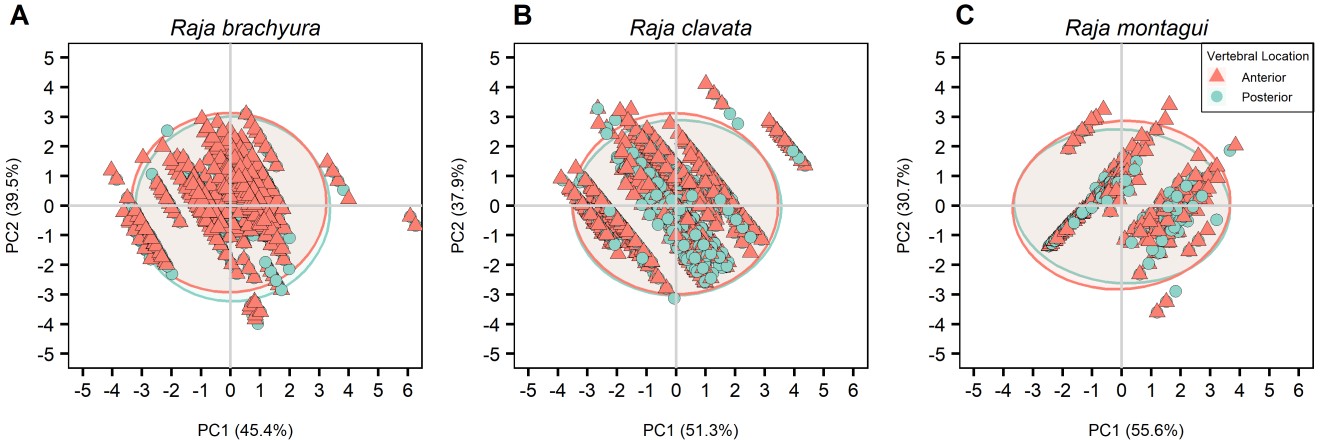

**Fig 8. PCA plot visually representing differences between vertebral location for the three species (A) *Raja brachyura*, (B) *Raja clavata,* and (C) *Raja montagui*.** Ellipses show 95% confidence intervals.

development of fish near their maximum size may account for the lack of older individuals (>14 years) estimated from bands for the species sampled in this study and previous research [55,56]. Francis, Campana and Jones [45] made the observation for tope shark (*Galeorhinus galeus*) which had a maximum estimated age of ~20 years as determined from sectioned vertebrae, and ~14 years from whole vertebrae [42] but a tagging study shows the species can live to at least 54 years [75,76]. Despite the differences in age estimations from each vertebral location in this study, this had little effect on the resulting growth curves.

**4.1.1. Vertebral location and growth curves.** Growth parameters estimated from whole stained vertebrae from both vertebral locations differed in *R. clavata* and *R. montagui* where posterior vertebrae overestimated the species growth coefficient while underestimating the asymptotic length. Statistical significance of growth curves between anterior and posterior vertebrae was only detected for *R. clavata* (Table 3; Fig 6B). Although *R. brachyura* presented an age composition similar to *R. clavata*, the growth curves obtained for anterior and posterior vertebrae were similar (p = 0.35; Chen test) (Table 3; Fig 6A). This result might indicate that posterior vertebrae could be considered as a reliable alternative to anterior ones. However, given the better ageing precision and reproducibility results, the discrepancies of ageing along the vertebral column, and that the majority of published studies utilise anterior vertebrae (either whole or sectioned), using posterior vertebrae for this species should be avoided in order to produce highly comparable results.

Despite this, the present study estimated higher growth coefficients using whole stained vertebrae for *R. brachyura* with respect to the growth parameters available in the English Channel [55], Irish Sea [20,57,77], while being more similar to other specimens from the same area [56]. *R. clavata*, produced growth parameters similar to those estimated in the English Channel [55], Irish Sea [57,77], and Welsh waters [53]. Finally, higher growth patterns were also estimated for *R. montagui* compared to what previously reported for the species in Northeast Atlantic [54,57,77], while being more comparable to values from the Irish Sea [20].

The differences between the growth curves estimated in the present study and previous studies could be ascribed to the different environmental [78,79] or fishing pressure [80] conditions present in the investigated areas. However, when comparing our data to other research, the relatively small sample size (especially for *R. montagui*) and the involvement of 8 distinct readers with varying levels of experience should be considered. As a result, the growth modelling results reported here for the purpose of evaluating the dependability of anterior and posterior vertebrae should be regarded as preliminary.

**Table 6. PCA values calculated concerning vertebral structure of anterior, stained vertebrae for each of the three skate species *Raja brachyura*, *Raja clavata* and *Raja montagui*.**

| Species | Variable | PC1 | PC2 |
|---|---|---|---|
| *Raja brachyura* | | R | R |
| | Age | −0.300*** | 0.794*** |
| | Total Length | −0.512*** | 0.615*** |
| | Longitude | 0.899*** | 0.294*** |
| | Latitude | 0.891*** | 0.323*** |
| | | R² | R² |
| | Sex | 0.013** | – |
| | Reader experience | – | 0.013* |
| | Preparation method | – | – |
| | Vertebral Structure | – | – |
| *Raja clavata* | | R | R |
| | Age | −0.500*** | 0.750*** |
| | Total Length | −0.730*** | 0.308*** |
| | Longitude | 0.817*** | 0.312*** |
| | Latitude | 0.720*** | 0.478*** |
| | | R² | R² |
| | Sex | 0.182*** | – |
| | Reader experience | – | 0.076*** |
| | Preparation method | – | – |
| | Vertebral Structure | – | – |
| *Raja montagui* | | R | R |
| | Age | 0.726 | −0.552 |
| | Total Length | 0.843 | −0.398 |
| | Longitude | 0.893 | 0.314 |
| | Latitude | 0.646 | 0.707 |
| | | R² | R² |
| | Sex | 0.054*** | – |
| | Reader experience | – | – |
| | Preparation method | 0.029** | – |
| | Vertebral Structure | 0.029** | – |

PCA values of Pearson correlation coefficients (R) of the quantitative variables with respect to the first (PC1) and second (PC2) principal components, and the proportion of the variance explained by the supplementary quantitative variables ($R^2$) for each PC. Significance is indicated where p-values are <0.05*, <0.01**, or < 0.001***.

## 4.2. Vertebral structure

In the present study whole and sectioned (anterior) vertebrae resulted in similar age estimations for *R. clavata* and *R. montagui*, but sectioned underestimated age in *R. brachyura*. When concerning posterior vertebrae, sectioned vertebrae consistently underestimated age in all species (Fig 4) emphasising the unreliability of using posterior vertebrae.

Previous investigations into comparing age estimations from different vertebral structures found mixed results between whole and sectioned vertebrae. In *P. glauca*, no significant differences were found in the age estimations between the two structural preparation methods [38]. Whereas another study concerning *H. portusjacksoni* reported that whole vertebrae

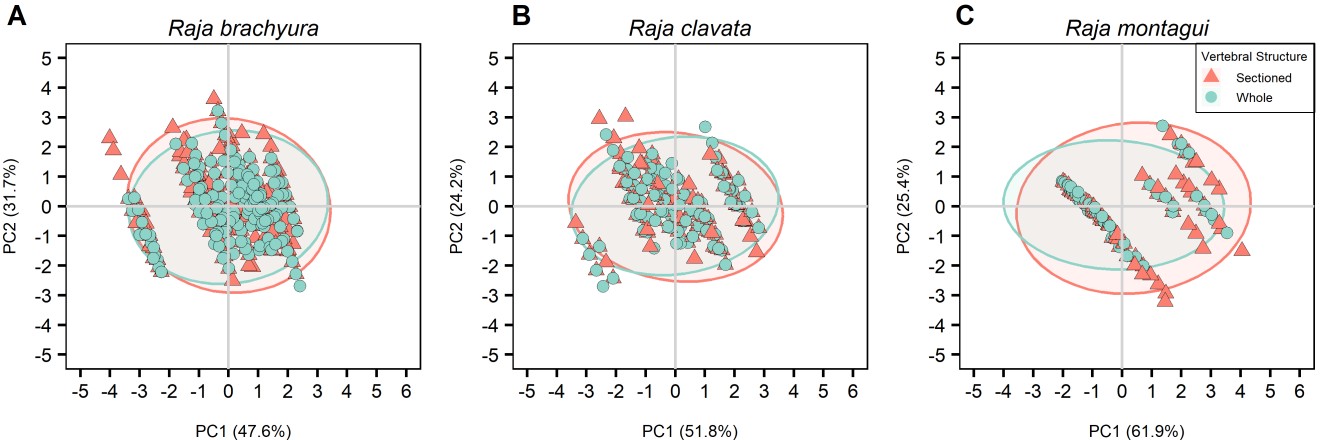

**Fig 9. PCA plot visually representing differences between vertebral structure for the three species (A)** *Raja brachyura*, **(B)** *Raja clavata,* and **(C)** *Raja montagui.* Ellipses show 95% confidence intervals.

produced lower age estimates and higher CV values compared to sectioned [37]. Similarly for *R. clavata* in the Adriatic Sea, whole vertebrae also produced higher CV and lower PA values [29].

High variability in aging precision was observed, which is common in elasmobranch species [81] and comparable to those of previous studies with a large number of age readers with different expertise [56]. Despite this, precision measures (APE, CV, and PA) indicated greater consistency and reliability for anterior, whole, stained compared to sectioned vertebrae for all species (Table 2). This is in contrast to a previous study where sectioning of *R. clavata* vertebrae increased overall precision [29]. Discrepancies between the two studies may stem from both methodological and environmental factors. A key difference is the staining method, where vertebrae were stained with cobalt nitrate and ammonium sulfide and was only applied to whole vertebrae, which limits comparability with our study. Additionally, each study had differing technical expertise; whole structures were prepared with more experience in this study while the previous study had more experience with preparing sectioned structures. This may have introduced bias, resulting in better quality preparation for the structure type each research team had more experience. The success of preparation techniques is influenced by vertebral cartilage biogeochemistry [82], which varies with environmental conditions [32,33]. Conditions in our study, conducted in the North Sea and English Channel, differ from conditions in the Adriatic Sea in the previous study [29], particularly with respect to temperature and salinity. There are also dietary differences between the populations of *R. clavata* in the North Sea [83] and the Adriatic Sea [84]. These differences in diet and environmental factors are expected to influence the vertebral cartilage biogeochemistry [32,33], thereby affecting the success of the preparation method used in different regions. Regardless, the results of our study suggest that sectioned vertebrae (stained or unstained) do not produce vertebral images that improve age readability over whole vertebrae.

### 4.3. Vertebral staining

A variety of staining methods are often used on vertebral structures to improve clarity and readability of band pairs [85–89]. However, stains or dyes are not always applied and may not always enhance growth band visibility for age reading [6,25,27,28] and tend to be species-specific [13,61]. This study found that staining was not always successful. In some cases, traces of the staining solution remained on the vertebrae making it more difficult to differentiate band pairs. Despite this, staining had little effect (positive or negative) on age reading precision and readability for vertebrae regardless of vertebral location or structure for all three species (Table 1). Future studies may take advantage of emerging techniques

**Table 7. PCA values calculated concerning vertebral staining of anterior, whole vertebrae for each of the three skate species *Raja brachyura*, *Raja clavata* and *Raja montagui*.**

| Species | Variable | PC1 | PC2 |
|---|---|---|---|
| *Raja brachyura* | | R | R |
| | Age | −0.082 | 0.996*** |
| | Total Length | −0.689*** | 0.005 |
| | Longitude | 0.949*** | 0.056 |
| | Latitude | 0.962*** | 0.035 |
| | | R² | R² |
| | Sex | 0.201*** | 0.009* |
| | Reader experience | – | 0.022** |
| | Preparation method | – | 0.143*** |
| | Vertebral Staining | – | 0.077*** |
| *Raja clavata* | | R | R |
| | Age | −0.586*** | 0.666*** |
| | Total Length | −0.618*** | 0.638*** |
| | Longitude | 0.845*** | 0.530*** |
| | Latitude | 0.930*** | 0.362*** |
| | | R² | R² |
| | Sex | 0.079*** | 0.165*** |
| | Reader experience | – | – |
| | Preparation method | – | – |
| | Vertebral Staining | – | – |
| *Raja montagui* | | R | R |
| | Age | 0.9111*** | 0.113 |
| | Total Length | 0.866*** | 0.383*** |
| | Longitude | −0.032 | 0.934*** |
| | Latitude | 0.564*** | −0.716*** |
| | | R² | R² |
| | Sex | – | 0.294*** |
| | Reader experience | – | – |
| | Preparation method | – | – |
| | Vertebral Staining | – | – |

PCA values of Pearson correlation coefficients (R) of the quantitative variables with respect to the first (PC1) and second (PC2) principal components, and the proportion of the variance explained by the supplementary quantitative variables (R²) for each PC. Significance is indicated where p-values are $<0.05$*, $<0.01$**, or $< 0.001$***.

such as using elemental analysis including laser ablation inductively coupled plasma mass spectrometry [27,90–92] and near-infrared spectroscopy [93,94] to corroborate age readings from vertebrae.

## 4.4. Management implications

The development of accurate and reliable ageing techniques has significant potential for improving sustainable management practices for skates and rays. The scientific advice and final decisions on the catches of the species currently disregard a number of key life-history characteristics, such as growth rates, longevity, and reproductive patterns. Knowledge of these life history characteristics are essential for more accurately assessing stock health and setting catch limits that reflect the true status

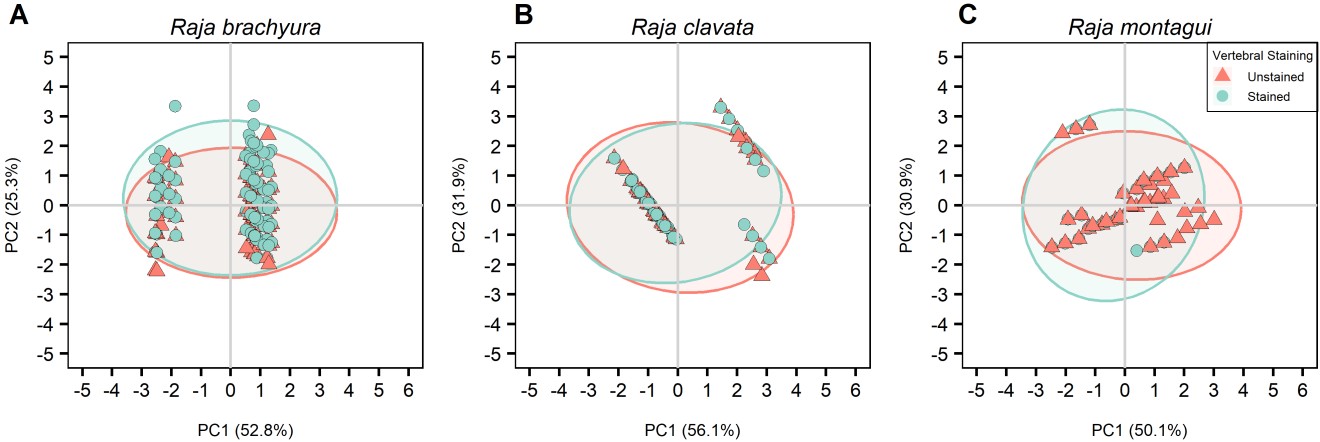

**Fig 10. PCA plot visually representing differences between vertebral staining for the three species (A)** *Raja brachyura*, **(B)** *Raja clavata*, **and (C)** *Raja montagui*. Ellipses show 95% confidence intervals.

of populations. Improved ageing methods enable more precise and species-specific stock assessments, which are crucial for moving away from less effective group-based management tools, such as group-TACs, which have been shown to poorly manage species with varying life histories and vulnerabilities like skates and rays [52]. By providing accurate data on species' age structures, age estimation supports the development of tailored management strategies that ensure catch limits and conservation measures are aligned with the actual status of the population, moving beyond broad, generalized approaches.

## 5. Conclusion

This study addressed some of the key issues regarding ageing of *Raja* species. The findings suggest that anterior vertebrae as the preferred locality for age reading due to their higher precision and potential discrepancies which arise from ageing posterior vertebrae. The results from this study showed increased precision for whole vertebrae. Staining had no effect on age reading precision suggesting this practice is not necessary for *Raja* species. In summary, the preparation method involving the use of anterior, whole, unstained vertebrae proved to be the most precise this study, offering significant reductions in both preparation time and cost by eliminating the need for sectioning or staining.

## Acknowledgments

We give thanks to Marcel de Vries, Hans Tap, Thomas Smith, Michiel Dammers, Ralf van Hal, Ingeborg de Boois, and other staff at Wageningen Marine Research for helping to collect samples. We give special thanks to Remko Verspui, Willie van Emmerik, and Gerard de Laak from Sportvisserij Nederland, and to Samara Hutting, Haniswita Haniswita, Agustin Capriati, Benrilo Mubarok Asdin, Dr. Sarah G Monic, Timo Michael Staeudle, Esther Nijkamp, Marwa Ahmed, Peter Zuther, Troy Spekking, Logan Binch, and Eliza Syropoulou for helping with dissections. We also thank the students of Wageningen University Ailynn Swiers, Saskia Aartsen, Renée van Embden, Jeroen Smit, Tanno van der Linden, and Louise Wijnands, for helping with dissections and reviewing literature during the research.

## Author contributions

**Conceptualization:** Eleanor S.I. Greenway, Jurgen Batsleer.

**Data curation:** Eleanor S.I. Greenway, Lorenzo L. Elias, Andrea Bellodi, Blondine Agus, Manfredi Madia, Mauro Sinopoli, Michele Palmisano, Ilse Maertens.

**Formal analysis:** Eleanor S.I. Greenway, Antonella Consiglio, Andrea Bellodi.

**Funding acquisition:** Jurgen Batsleer.

**Investigation:** Eleanor S.I. Greenway.

**Methodology:** Eleanor S.I. Greenway.

**Project administration:** Eleanor S.I. Greenway, Jurgen Batsleer, Jan Jaap Poos.

**Resources:** Eleanor S.I. Greenway, Jurgen Batsleer, Jan Jaap Poos.

**Supervision:** Jurgen Batsleer, Jan Jaap Poos.

**Visualization:** Eleanor S.I. Greenway, Antonella Consiglio, Andrea Bellodi, Pierluigi Carbonara, Mauro Sinopoli.

**Writing – original draft:** Eleanor S.I. Greenway, Lorenzo L. Elias, Antonella Consiglio, Andrea Bellodi, Jurgen Batsleer.

**Writing – review & editing:** Eleanor S.I. Greenway, Lorenzo L. Elias, Antonella Consiglio, Andrea Bellodi, Blondine Agus, Jurgen Batsleer, Karen Bekaert, Pierluigi Carbonara, Manfredi Madia, Mauro Sinopoli, Michele Palmisano, Ilse Maertens, Jan Jaap Poos.

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
