## [Decision Letter · Decision Letter 0]

8 Sep 2025

*Raja brachyura* , *Raja clavata* , and *Raja montagui*

Dear Dr. Greenway,

We look forward to receiving your revised manuscript.

Kind regards,

Joel Harrison Gayford

Academic Editor

PLOS ONE

Journal Requirements:

4. We note that Figure 1 in your submission contain map/satellite images which may be copyrighted. All PLOS content is published under the Creative Commons Attribution License (CC BY 4.0), which means that the manuscript, images, and Supporting Information files will be freely available online, and any third party is permitted to access, download, copy, distribute, and use these materials in any way, even commercially, with proper attribution. For these reasons, we cannot publish previously copyrighted maps or satellite images created using proprietary data, such as Google software (Google Maps, Street View, and Earth). For more information, see our copyright guidelines: http://journals.plos.org/plosone/s/licenses-and-copyright.

Reviewers' comments:

Reviewer's Responses to Questions

**Comments to the Author**

1. Is the manuscript technically sound, and do the data support the conclusions?

Reviewer #1: Yes

Reviewer #2: No

2. Has the statistical analysis been performed appropriately and rigorously?

Reviewer #1: Yes

Reviewer #2: No

3. Have the authors made all data underlying the findings in their manuscript fully available?

Reviewer #1: Yes

Reviewer #2: Yes

4. Is the manuscript presented in an intelligible fashion and written in standard English?

Reviewer #1: Yes

Reviewer #2: Yes

Reviewer #1: In general, this is a very strong manuscript. The methods were very rigorous and thoroughly explained so that anyone may recreate this procedure. I have a few minor recommendations that I believe would contextualize the need for age validation in elasmobranchs, but the methods and results are the focus of this study.

Intro: I would like to see more detail in this section on the reasons why age reading success is species/diet/environmentally dependent and what aspects of life history makes it difficult to have a sweeping methodologically across the taxa. Moreover, why is ageing such an important skill to hone for elasmobranchs? Both of these points are addressed in the discussion, but I would like to see some hint of them in the introduction so that I am drawn into the importance of this study.

L44 I think this sentence can be rewritten to be stronger ("fish" is used twice)

L50 give indication of why those structures are used in elasmobranchs (calcified, records age similar to otoliths); move parentheses to end of sentence ("sharks, skates and rays (elasmobranchs)"

L57-8 This sentence feels out of place here wince we don't have context for this particular species, but the inclusion of a specific study and species is something that should be done in the introduction.

L59-61 More details on why the authors couldn't validate

L61 Unclear what the first discrepancy is, other than some papers invalidated vertebral growth

L61-4 This sentence and first half of the second sentence can be combined

L65 Provide examples of environmental factors (temperature? salinity?)

L77 More info on these "other studies" - species? underestimation? etc

Methods: Really strong section, very thorough and clearly explained

L96-7 This sentence should be start of results

L109-111 I think this sentence is missing some punctuation

L111-12 Also put this in results

L150 Eight readers at each institute or across all three?

Results: Thorough. Would benefit from in text reported test statistics and values/means.

L223 "on average" a statistical average? Is it significant?

Table 2 "Modal age range" is slightly confusing. I think it means the age range of the whole species sample as determined by the modal age of each individual across all reads, but it also could be the extent of the range in age read from the readers. Since it's already established that age was determined using mode read, "modal" can be removed from the term. Also, this needs units (years)

L234 "observed" implies the animal was monitored since birth and is thus validated to be 6. "estimated" would be more appropriate, since the true age is never known unless the animal was born and kept in captivity.

Sections 3.2-3.3 These sections are missing reported test statistics (especially p values) and values of what is being compared..

L282 "each both" is redundant

Discussion: Great analysis and inclusion of relevant studies/literature. Overall strong.

L346-7 Start discussion with your findings; are elasmobranchs understudied? That wasn't mentioned in the introduction; I recommend a broader opening paragraph where a recommendation of best method is made, like last sentence of abstract and in the conclusion.

L357-61 Great analysis!

L370 "observed" -> "estimated"

L371 The species doesn't live to 54, individuals in the species can live to 54

L372 "a minimum of 54 years" sounds like all tope sharks must live to 54. Change to "can live to at least"

L379 This sentence says the same thing... why would it say "although" if the age composition and growth curve are both similar in the two species?

L380 Give p value, not "no significant difference"

L391 Sentence end?

L394-5 This is a fairly large sample size for elasmobranchs, but not as much compared to teleosts

399 Start section 4.2 with this study's findings on vertebral structure. The beginning of every section should start with restating the relevant findings from the results (lines 404-408)

409 It's not expected, but it is common

L411 "APE and CV were lower, and in general PA was higher" This sentence also appears in line 222, and I'm confused by it again. It would help to give values with these types of statements such as APE and CV were x% lower and 75% of PA was higher than... In general, it's best to avoid saying "in general" (I couldn't resist)

415 There's an extra "vertebrae" in this sentence

412-29 Really great analysis and discussion here

L437 This is a really great and valuable section, but I would have liked to see it introduced in the introduction so that the reader has an understanding of why it is important to improve age studies in elasmobranchs

Conclusion: Good, but maybe not necessary as it's own section. If the last sentence is removed, it can be added to the discussion. Up to the authors, but I don't think this text needs to be on its own.

L456 Change "the authors suggest" to "the findings suggest" or "the findings point to" so that the suggestion is driven by data, not the authors' interpretation

L462-3 Make into one paragraph?

L465-7 Unnecessary sentence

Reviewer #2: General Comment to Authors

I like the concept of this study and I am always encouraged to read manuscripts seeking to improve the problematic field of aging. However, I have concerns some content, methodological aspects, and elements lacking in the discussion.

I make 5 over arching comments, which are unpacked in more detail in line comments below. I encourage the authors to think about the framing of this study, as there are some really good elements here. I am just not convinced that the method has resulted in a recommendation that is going to lead to the best estimates of true age in future work; only the most statistically constrained estimates in the context of reader agreement, and whether this same result would be achievable in a sample including longer lived species is questionable.

1. The introduction does not do a good job of situating the current ‘best practice’ of preparing and reading band pairs in vertebrae. The convention is to section anterior vertebrae as this generally results in higher age estimations; important for a field prone to systematic under aging through vertebral analysis, and convention since 2004 (Cailliet and Goldman 2004).

2. It is not explained how the authors identified the birth mark between methods.

3. The use of taking the ‘modal age’ in lui of consensus reads among agers is prone to introducing systematic under ageing bias – what if the oldest age estimates were most true of actual age? That consideration cannot be made within the current methodology. The variation on age estimates (fig 4 & 5) is quite large and only emphasizes my concern. The present method selects for comfortable age-bias/precision stats, but does not necessarily ensure that the most true/best/pre cautionary final age estimations have been made in reality.

4. The modeling approach of length at age data is quite antique. Its pretty firmly accepted to use a multi-model approach (not just a priori selection of the von bertalanffy function, Smart et al 2016)), and better yet a Bayesian model where priors of readily available information on size-at-birth/hatching and maximum size can be used to constrain the parameters of L0 and Linf (Smart and Grammar 2021).

5. My final comment extending from 429 argues that this study has selected the best preparation method for vertebrae to improve age bias/precision metrics, not necessarily the method that would produce the most accurate estimation of true age. The concept of counting whole vs section vertebrae is again quite antiquated and really, its the bias in younger age classes in this study which is most likely what has resulted in comparable age estimates between whole and sectioned vertebrae. It’s a dangerous assertion to encourage use of whole vertebrae for skate on the back of this work, as it will not be appropriate for individuals attaining older ages than the present studies sample size. This is known from over 40 years of age and growth research, and goes against best practice guidelines of research in the field that has led to the presently accepted conventions on vertebrae preparation for ageing.

Line Comments

Line 51

Add caudal thorns to this list.

Line 54

Reframe this paragraph; staining is not always used, and to my awareness its probably more common not to stain. There are studies that have discussed the relative null effect of staining, and these should be referred to here. See for example Burke et al. 2020, which subsequently cites some useful papers exploring the use of staining; along with others cited in the present MS. The other key point to note with staining (more for discussion) is that this technique was popular before photographic manipulation was possible. These days, its probably more effective to play around with brightness and contrast of images to achieve the same effect as staining – this is why staining is hardly used now; digital advancements have made it largely superfluous. Along with this is the development of techniques like XRF elemental mapping, where calcium and phosphate (the most abundant elements) can be discreetly viewed along the transect.

Burke, P. J., Raoult, V., Natanson, L. J., Murphy, T. D., Peddemors, V. & Williamson, J. E. (2020). Struggling with age: Common sawsharks (Pristiophorus cirratus) defy age determination using a range of traditional methods. Fisheries Research 231, 105706.

I also think it needs to be clear that vertebrae are most commonly sectioned, as previous work has identified under aging of whole centra;

Discussed a bit in following + other papers already cited in this MS. This introduction doesn’t really set the appropriate context for accepted methodological conventions in the field that are designed to improve readability and minimise under aging. This is important context to this present work – the convention is to section anterior vertebrae; this study largely seeks to validate or invalidate this convention for these species of skate.

Harry, A. V. (2018). Evidence for systemic age underestimation in shark and ray ageing studies. Fish and Fisheries 19, 185-200.

Cortés, E., Grant, M. I., Dureuil, M. & Ellis, J. R. (2024). Life history characteristics. In: R.W. Jabado, A.Z.A. Morata, R. Bennett, B. Finucci, J. Ellis, S. Fowler, M.I. Grant, A.P.B. Martins and S. Sinclair (Eds.), The Global Status of Sharks, Rays, and Chimaeras, pp. chapter 3, 33–38. IUCN/SSC Shark Specialist Group. IUCN, Gland, Switzerland and Cambridge, UK

Goldman, K. J., Cailliet, G. M., Andrews, A. H. & Natanson, L. J. (2012). Assessing the age and growth of chondrichthyan fishes. In Biology of Sharks and Their Relatives (Carrier JC, Musick JA & MR, H., eds.), p. 423. Boca Raton, FL: Taylor & Francis Group.

MacNeil, M. & Campana, S. (2002). Comparison of whole and sectioned vertebrae for determining the age of young blue shark (Prionace glauca). Journal of Northwest Atlantic Fishery Science 30, 77-82.

Line 64

The biochemical structure of vertebrae (phosphate hydroxyapatite [Ca10(PO4)6(OH)2]) is not variable between species. Its density, flexibility are influenced most likely by species morphology and stress forces needed in skeletal structure owing to swimming mode and ecological activity (along the spectrum of sedentary to highly mobile- if a species is swimming a lot, it needs a string skeleton to anchor large muscles to). Here, diet and other environmental factors are referred to, This releates to trace element incorporation in the Ca10(PO4)6(OH)2 matrix or elements entrapped in the protein matrix. These factors would only have minor relevance to structural properties (density, hardness, etc) of vertebrae in the context the authors are talking about here. These needs some revision to better describe structural properties of vertebrae, there composition, and what factors influence composition. Lisa Natanson explored these concepts late in her career;

James, K. C. & Natanson, L. J. (2020). Positional and ontogenetic variation in vertebral centra morphology in five batoid species. Marine and Freshwater Research 72, 887-898.

Natanson, L. J., Skomal, G. B., Hoffmann, S. L., Porter, M. E., Goldman, K. J. & Serra, D. (2018). Age and growth of sharks: do vertebral band pairs record age? Marine and Freshwater Research 69, 1440-1452.

Natanson, L., Andrews, A., Passerotti, M. & Wintner, S. (2018). History and Mystery of Age and Growth Studies in Elasmobranchs: Common Methods and Room for Improvement. In Carrier, J. C., Heithaus, M. R., & Simpfendorfer, C. A. (Eds.). Shark research : Emerging technologies and applications for the field and laboratory, pp. 177-194. Taylor and Francis Group. pp. 178-194.

However, I wonder how relevant this really is to the present paper? It hasn’t explored how varied vertebral structures hold up against vertebral preparation techniques. Better to set the context of conventional methods for vertebral preparation for ageing, and why it is we have arrived at the present convention of sectioning anterior vertebrae, where staining has limited benefit. That’s the state of knowledge this MS needs to build a hypothesis of that would improve the field.

Line 154

Please explain the process of identifying the birthmark for whole and sectioned vertebrae. Its typical to get a translucent band at the birthmark, so did counts begin from the second translucent band in whole vertebrae? Another advantage to sectioned vertebrae is that the birth mark is more easily identified, as in addition to a thin translucent band that is typically (but not always) laid, is that there will usually be an acute angle change in the intermedalia at the birth mark, and sometimes a visible ‘notch’. These characters cannot be identified in whole vertebrae, so how was the birth mark identified? Was the accuracy of its identification corroborated with sectioned vertebrae? One of the major error sources of age estimation is mis identification of the birth mark, and this study has not explained how it was identified, and what constituted a growth band there after. It just says growth bands were counted.

Line 187

It’s a bit of an old fashioned parametrization of the growth model used here. The parametrization that includes L0 is more useful, as L0 serves as a proxy for size at birth, with t0 has no biological meaning. The use of the parametrization with L0 also allows Bayesian A&G models to be used, and L0 and Linf can more easly have informed priors, based on knowledge of birth size and maximum size. Its probably not possible to accurate inform a prior of t0. Its not essential, but I would be highly encouraging of considering use of a more contemporary approach to modelling the length at age data

Please see

Smart, J. J., Chin, A., Tobin, A. J. & Simpfendorfer, C. A. (2016). Multimodel approaches in shark and ray growth studies: strengths, weaknesses and the future. Fish and Fisheries 17, 955-971.

Smart, J. J. & Grammer, G. L. (2021). Modernising fish and shark growth curves with Bayesian length-at-age models. PLoS One 16, e0246734. (includes very intuitive R package)

This section also needs to be clear that the a priori selected model is the von bertalanffy growth function. Again, see Smart et al 2016 above, as this strays from contemporary approaches to modelling length at age data whereby a multimodel approach is encouraged as best practice.

Line 429 and then general discussion remarks

Yes, but only for the young spectrum of age estimates encountered in this study. This discussion is crying out for explicit mention that a key constraint here is the young ages encountered <8. The key advantage of sectioning comes when highly compressed ‘older age bands’ need to be examined, when animals have 20, 30, 40 band pairs to count. It is not really all that surprising that whole vertebrae performed equal, or slightly better than section vertebrae I some instances for these young age classes where growth bands are inherently larger and more easy to count – the need for sectioning was not born out of difficulties counting bands in younger age classes, it was from difficulties encountered in counting bands in older age classes, which are not included in this study. Further, is there any insight to offer in where the larger age discrepancies in sectioned vertebrae arose? Was it birth mark, distal edge? That would be useful information; identifying vertebral zones leading to mis counts among readers.

The other factor here, is the assumption is made that band pairs equate to annual growth increments. There was more variability in the sectioned vertebrae, but how do you know that those age estimates were not better than whole vertebrae? By not sectioning, smaller more indiscreet bands are hard to identify, and may have been missed. Our built ‘best practice’ of ageing calls for sectioning for this reason (e.g., Cailliet and Goldmans catelouge of papers on this topic; built from over 30 years of examining ageing studies and trailing methods; along with prior mentioned better ability to identify the birth mark; Cailliet and Goldman 2004). What this study has done is identify the best preparation method of vertebrae from these species of skate, in these ‘age’ classes, that allows a wide range of readers to most commonly agree on their ‘age estimates’ and thereby produce the best age bias/precision stats. Whether underaging has occurred is an un answered question. An interesting thing here, and counter to most literature, is that higher age estimates were made for R. brachyura with whole vertebrae, but not for the other two species. Why do you think this is? Have whole vertebrae resulted in better age-bias stats because ambiguous bands, or smaller bands are less clear in whole vertebrae compared to section vertebrae so that there is effectively less source for error ? – have whole vertebrae potentially missed some smaller bands that could well represent annual depositions under the assumption band pairs are accreted annually for these species? If there are less banding structures to count, age-bias stats are inherently going to be lower

Line 237 states that sectioned vertebrae had higher ‘modal’ age estimates for R. clavate and R. montagui, but, these precision and bias stats were higher. Given that aging is about producing estimates that most reflect true age of the animal so that those ages can be fed into management initiatives, is it better to have higher more precautious ages with slightly worse age bias/precision stats, or lower more conservative ages with neater age bias/precision stats? Its an partly uncomfortable question that this study has raised, and it needs to be addressed. The goal of aging is not low APEs, its proving the best advice to management of growth of the species, and given we know underestimation occurs from vertebral ageing, it seems to make sense to use the precautionary principle and err toward older estimates, even if it means lower agreement among readers. Better to over age than under age.

Furthermore, the concept of selecting the ‘modal age’ is not something I entirely agree with – how to you know that the oldest age estimate wasn’t the more accurate to true age and that the modal age selection is not a form of methodological bias leading to systematic age underestimation? The variation on age estimates (fig 4 & 5) is quite large and only emphasizes my concern. Where any consensus reads done? (if not, again straying from conventional methodology), particularly with the ager who made the oldest age estimation for each individual so that they could make their case as to their decision to count the band structures they could identify? In a field where underage estimation is an issue, again, it’s a backward facing decision in respect to the history of this problematic field. The present method selects for comfortable age-bias/precision stats, but does not necessarily ensure that the most true/best/pre cautionary final age estimations have been made in reality. As such, the title is not entirely appropriate; it should be ‘Vertebral preparation to improve band pair identification agreement among readers’ or something to that effect.

These technicalities need some discussion. The field of ageing with vertebrae is plagued with inferences, and sometimes proven examples of systematic underestimation of age (e.g., the MRC study referred to for Tope, theres others for white sharks, where if we believed whole vertebrae estimates they’d have a max age of about 20, not 70+). My over arching point is that whole vertebrae may have resulted in clean APE’s and CV’s, but have they resulted in more accurate estimations of the age of these individuals – this is a separate point to the number of readers that agreed on an age, as that will err toward conservative counts. This discussion needs to consider these aspects – do whole vertebrae improve age estimates or just age precision metrics among readers? My view is that the latter is probably true and this discussion needs to be framed to answering this question as it’s the over arching unknown to come out of this study.

Some good well presented and discussed points are that anterior vertebrae resulted in higher age estimates generally, and were ‘more readable’ and that staining had very limited effect. This is corroborated in the literature, and this study adds to the evidence base of these factors. I have no issues with the PCA parts of this study, though I wonder if there is really enough latitudinal and longitudinal scope to provide robust insight. I’d also be interested to see the effects of reader experience on ages

Figure 2, please mark what was identified as the birth mark on this figure, and it would be useful to mark the identified band pairs counted also (if possible considering the modal age method). These images lack meaning otherwise for the reader

**Do you want your identity to be public for this peer review?** For information about this choice, including consent withdrawal, please see our Privacy Policy

Reviewer #1: No

Reviewer #2: No

---

## [Author Response · Author response to Decision Letter 1]

23 Oct 2025

Comments from the Editor

Journal Requirements:

The manuscript style requirements were double checked and changes were made to match the example document where necessary.

These methods of sampling in this study did not involve any additional mortality outside of standard fishing practices and researchers had no involvement of killing fish used in this study. Therefore, this research did not require any animal experimentation permits. The following text has been added, line 121 – 127:

“Individuals of R. brachyura, R. clavata, and R. montagui were collected between June 2018 and March 2024. Samples and data were collected either from landed individuals intended for human consumption, or as part of routine data collection in a commercial fisheries program for discard sampling in The Netherlands. This study was non-experimental as samples obtained from discard sampling involved opportunistically collecting individuals that were already deceased, where The Dutch Experiments on Animals Act does not apply. An ethical review by the Statement Animal Experiment Committee was therefore not required. No additional mortality or animal discomfort beyond standard fishing operations was caused for sample collection for the purpose of this study.”

The data were under review in the DANS data repository and are now publicly available with the doi link as stated in the submission process: https://doi.org/10.17026/LS/ARNWEH.

4. We note that Figure 1 in your submission contain map/satellite images which may be copyrighted. All PLOS content is published under the Creative Commons Attribution License (CC BY 4.0), which means that the manuscript, images, and Supporting Information files will be freely available online, and any third party is permitted to access, download, copy, distribute, and use these materials in any way, even commercially, with proper attribution. For these reasons, we cannot publish previously copyrighted maps or satellite images created using proprietary data, such as Google software (Google Maps, Street View, and Earth). For more information, see our copyright guidelines: http://journals.plos.org/plosone/s/licenses-and-copyright.

Figure 1 was created using the “rnaturalearth” package in R studio and is not a copyrighted image. This is now clarified in the figure legend with the following text “...Locations were estimated from the national fisheries catch and effort database by linking fish landings with fishing trip locations, and visualised using the “rnaturalearth” package in R version 4.4.1.”

Reviewer #1 Comments:

1. L44 I think this sentence can be rewritten to be stronger ("fish" is used twice)

Changed as suggested

2. L50 give indication of why those structures are used in elasmobranchs (calcified, records age similar to otoliths); move parentheses to end of sentence ("sharks, skates and rays (elasmobranchs)"

Parentheses have been moved as suggested and the sentence has been changed to: ”Alternative calcified structures, such as spines, rayfins, caudal thorns, and vertebrae, often feature patterns that reflect annual growth bands and are therefore used for age estimation in these species [6, 7].”

3. L57-8 This sentence feels out of place here wince we don't have context for this particular species, but the inclusion of a specific study and species is something that should be done in the introduction.

We agree with the reviewer and the sentence has been adapted and moved towards the end of the introduction.

4. L59-61 More details on why the authors couldn't validate

We agree and now clarify why vertebral age validation can fail across elasmobranchs. Validation is often unachievable when growth bands exhibit inherently poor optical contrast and frequent accessory checks features that are species-specific and linked to vertebral cartilage biogeochemistry. In addition, band deposition can be non-annual or temporally variable among taxa and environments, and external validation approaches (e.g., oxytetracycline mark–recapture, known-age specimens, bomb-radiocarbon) are not universally applicable due to ethical, logistical, or life-history constraints.

5. L61 Unclear what the first discrepancy is, other than some papers invalidated vertebral growth

Sentence has been changed to “Although age estimation in elasmobranchs has been validated in several studies [29-31], vertebral ageing cannot be universally confirmed because band visibility and periodicity vary among species; this discrepancy reflects interspecific differences in the biogeochemistry of vertebral cartilage, due to variation in matrix composition, degree of mineralization, and trace-element incorporation—factors influenced by diet and environmental conditions (e.g., temperature, salinity, dissolved oxygen, pH/carbonate chemistry, depth) [32, 33]. Moreover, the use of vertebrae for age estimation has been invalidated for several species, including the common sawshark (Pristiophorus cirratus) [34] and the Pacific angel shark (Squatina californica) [35]. As a result, a single standardized technique for age estimation across all elasmobranchs remains unlikely [6, 36]”

6. L61-4 This sentence and first half of the second sentence can be combined

Changed as suggested.

7. L65 Provide examples of environmental factors (temperature? salinity?)

We now provide concrete examples of environmental drivers (temperature, salinity, dissolved oxygen, pH/carbonate chemistry, depth), with citations.

8. L77 More info on these "other studies" - species? underestimation? Etc

We added more details from the previous studies as suggested: “However, other studies on different shark (Squatina dumeril, Carcharodon carcharias, Lamna nasus, Isurus oxyrinchus, Alopias vulpinus, Prionace glauca, and Carcharhinus obscurus) [41], skate, (Leucoraja erinacea, Leucoraja ocellata, Dipturus laevis), and ray species (Dasyatis sabina, and Urobatis halleri) [24] have reported age underestimations in posterior vertebrae as centrum morphology and band-pair counts vary along the vertebral column [24, 41]”

9. L96-7 This sentence should be start of results

Sentence adapted to the results as suggested.

10. L109-111 I think this sentence is missing some punctuation

This sentence has now been broken into two sentences.

11. L111-12 Also put this in results

This particular section is currently in the methods section because it serves as support for our choice in vertebrae selection, which is detailed in the subsequent sentence.

12. L150 Eight readers at each institute or across all three?

This was eight readers across all three, sentence rephrased to: “Age estimates were carried out independently by eight readers representing institutes in The Netherlands, Belgium, and Italy.”

13. L223 "on average" a statistical average? Is it significant?

Average replace with ‘in most cases’.

14. Table 2 "Modal age range" is slightly confusing. I think it means the age range of the whole species sample as determined by the modal age of each individual across all reads, but it also could be the extent of the range in age read from the readers. Since it's already established that age was determined using mode read, "modal" can be removed from the term. Also, this needs units (years)

We agree with the reviewer and the word modal has been removed while including the unit (yr). We also added more information to the figure legend for clarity: “Table 2. Summary of ageing precision results of each vertebral preparation method. The age range in the precision measures column reflects the age range of the modal ages as estimated by all age readers.”

15. L234 "observed" implies the animal was monitored since birth and is thus validated to be 6. "estimated" would be more appropriate, since the true age is never known unless the animal was born and kept in captivity.

Changed as suggested.

16. Sections 3.2-3.3 These sections are missing reported test statistics (especially p values) and values of what is being compared..

The authors regret this information was missing form the original text, significant p-values and test statistics are now given in the text. Non-significant results are given in the figures.

17. L282 "each both" is redundant

Corrected as suggested.

18. L346-7 Start discussion with your findings; are elasmobranchs understudied? That wasn't mentioned in the introduction; I recommend a broader opening paragraph where a recommendation of best method is made, like last sentence of abstract and in the conclusion.

We thank the reviewer for their comment and the paragraph in the beginning of the discussion has been changed as suggested.

19. L370 "observed" -> "estimated"

Corrected as suggested

20. L371 The species doesn't live to 54, individuals in the species can live to 54

Corrected as suggested

21. L372 "a minimum of 54 years" sounds like all tope sharks must live to 54. Change to "can live to at least"

Corrected as suggested

22. L379 This sentence says the same thing... why would it say "although" if the age composition and growth curve are both similar in the two species?

The sentence has been adjusted to improve clarity: “Statistical significance of growth curves between anterior and posterior vertebrae was only detected for R. clavata (Table 3; Fig 6B). Although R. brachyura presented an age composition similar to R. clavata, the growth curves obtained for anterior and posterior vertebrae were similar (p = 0.35; Chen test)”

23. L380 Give p value, not "no significant difference"

P-values were added as suggested.

24. L391 Sentence end?

Sentence corrected to ‘while being more comparable to values from the Irish Sea’

25. L394-5 This is a fairly large sample size for elasmobranchs, but not as much compared to teleosts

We added text to clarify that our sample size is particularly small for R. montagui, especially compared to teleost studies as the reviewer correctly identifies.

26. 399 Start section 4.2 with this study's findings on vertebral structure. The beginning of every section should start with restating the relevant findings from the results (lines 404-408)

Paragraph re-arranged as suggested.

27. 409 It's not expected, but it is common

Changed as suggested

28. L411 "APE and CV were lower, and in general PA was higher" This sentence also appears in line 222, and I'm confused by it again. It would help to give values with these types of statements such as APE and CV were x% lower and 75% of PA was higher than... In general, it's best to avoid saying "in general" (I couldn't resist):

Line 222 (now line 265) has been edited to include specific values derived from the table: “Both APE was lower by 2–19% and CV by 3–25% for anterior, whole, stained vertebrae compared to sectioned vertebrae for all species. PA was also consistently higher by 13–14% for whole compared to sectioned vertebrae. In most cases, anterior vertebrae resulted in CV values 1–17% lover compared to posterior vertebrae, regardless of vertebral structure or staining (Table 2).”

The discussion has also been adjusted to: “Despite this, precision measures (APE, CV, and PA) indicated greater consistency and reliability for anterior, whole, stained compared to sectioned vertebrae for all species (Table 2).”

In general, we agree with the reviewer that in general is it best to avoid ‘in general’.

29. 415 There's an extra "vertebrae" in this sentence

Deleted where necessary

30. L437 This is a really great and valuable section, but I would have liked to see it introduced in the introduction so that the reader has an understanding of why it is important to improve age studies in elasmobranchs

We thank the reviewer for their comment, we have now moved part of this section to the introduction.

31. Conclusion: Good, but maybe not necessary as it's own section. If the last sentence is removed, it can be added to the discussion. Up to the authors, but I don't think this text needs to be on its own.

We appreciate the constructive comment but we decide to keep the conclusion section separately.

32. L456 Change "the authors suggest" to "the findings suggest" or "the findings point to" so that the suggestion is driven by data, not the authors' interpretation

Changed as suggested.

33. L462-3 Make into one paragraph?

Changed as suggested.

34. L465-7 Unnecessary sentence

Deleted as suggested.

Reviewer #2 General Comment to Authors

1. The introduction does not do a good job of situating the current ‘best practice’ of preparing and reading band pairs in vertebrae. The convention is to section anterior vertebrae as this generally results in higher age estimations; important for a field prone to systematic under aging through vertebral analysis, and convention since 2004 (Cailliet and Goldman 2004).

While we appreciate the comment from the reviewer, in our view, the convention to section vertebrae is not as strong as suggested for Raja species which have recently been aged with whole vertebrae (Thys et al., 2023; Bellodi et al., 2024) and not unequivocally supported by previous observations. We have adapted the introduction which emphasises the uncertainty in both vertebral preparation methods.

“Comparisons of vertebral preparation methods have also shown variable results among elasmobranch species. Previous research shows that sectioned vertebrae produce higher and more accurate ages compared to whole vertebrae in as seen in the spot-tail shark (Carcharhinus sorrah), the Australian blacktip shark (Carcharhinus tilstoni) [36], and the Port Jackson shark (Heterodontus portusjacksoni) [37]. In contrast, other studies show that whole vertebrae produce similar ages to sectioned vertebrae in young blue shark (Prionace glauca) [38], or may even be preferred over sectioning vertebrae as in the crocodile shark (Pseudocarcharias kamoharai) [39]. Regardless, age underestimations occur in both whole and sectioned vertebrae in older individuals as growth bands either become tightly grouped on the vertebral edge making them more difficult to distinguish [40] or growth bands deposition halts once somatic growth slows or ceases [24, 41]. Such age underestimations have been validated from tagging studies [36, 42, 43], or bomb radiocarbon dating [44, 45]. Collectively, these studies highlight both the uncertainty and limitations of using vertebral centra for age estimations and demonstrate that preparation methods are highly species-specific. However, vertebral ageing appear most reliable fo

---

## [Decision Letter · Decision Letter 1]

17 Dec 2025

*Raja brachyura* , *Raja clavata* , and *Raja montagui*

Dear Dr. Greenway,

Thank you for submitting your manuscript to PLOS ONE. After careful consideration, we feel that it has merit but does not fully meet PLOS ONE’s publication criteria as it currently stands. Therefore, we invite you to submit a revised version of the manuscript that addresses the points raised during the review process.

We look forward to receiving your revised manuscript.

Kind regards,

Joel Harrison Gayford

Academic Editor

PLOS One

Journal Requirements:

Reviewers' comments:

Reviewer's Responses to Questions

**Comments to the Author**

Reviewer #2: (No Response)

2. Is the manuscript technically sound, and do the data support the conclusions?

Reviewer #2: Yes

3. Has the statistical analysis been performed appropriately and rigorously?

Reviewer #2: Yes

4. Have the authors made all data underlying the findings in their manuscript fully available?

Reviewer #2: Yes

5. Is the manuscript presented in an intelligible fashion and written in standard English?

Reviewer #2: (No Response)

Reviewer #2: I thank the authors for their response to my comments and improvements made to the manuscript. At heart, I do not agree with the modal age approach, but I am satisfied that the first paragraph of the discussion outlines the key constraint for the reader, and the purpose of its use herein. There is scope to justify the reasoning for this approach in the methods additionally, as it is a foundational decision made in the conduct of the methodology of this study, and an action that influences the results and their interpretation (see below comment for a suggestion, as you’ve actually provided a really good couple of sentences in a reply to a comment).

Considering this text now added to the start of the discussion, and reply to my comment 3., I wonder whether the title is entirely appropriate. The final sentence of the first discussion paragraph for example is correct, and also implies the title is misleading

“Therefore, our finding that anterior, whole, unstained vertebrae provided the highest reader precision and agreement should be interpreted within this context”

A more appropriate title as such would be a synonym of:

A comparison of vertebral preparation techniques to increase reader precision and agreement in vertebral band pair identification in northwest Atlantic skates.

This is what the study has done. I recommend saving the ‘age estimation’ for the follow up manuscript that you mention

A few responses to previous comment reply’s and just a few more line comments below. Well done to the authors, this was quite a large review and I appreciate the time in constructing considered responses.

Reply to comment 3.

The statement ‘modal age remains a well-established and widely accepted practice in age and growth validation studies’ is something I just do not consider to be true. Noting the authors list of referred to papers (Campana 2001; Goldman 2005; Vitale et al., 2021; ICES, 2022) in their response to comment 3., Campana 2001 for example speaks to ‘modal progression’ which is in the context of length class modes corresponding to ages, and as such, whether length modes in an annual breeding population can be used to estimate age from length; not taking the mode of age estimates across a range of readers who have examined banding patterns in calcified hard parts to determine age. I haven’t had much luck locating others cited, but ensure there is no confusion in ‘modal progression’ which is different to the present method. Regardless, better to justify the use of using modal ages to the readership more so than me. I like the justification provided in the first paragraph in the authors reply to the original comment 3., and would encourage its inclusion within the paragraph ending at line 194

‘…. modal age can potentially introduce a systematic underageing bias, particularly if the highest (oldest) age estimates more closely approximate true age. However, our primary objective was not to determine absolute accuracy but rather to assess how different preparation methods influence the readability and consistency of vertebral age estimates across multiple readers. The modal age approach enabled us to quantify reader consensus and exclude indeterminate cases (i.e., instances with no clear mode), thereby focusing the analysis on relative precision rather than absolute accuracy.’

Reply to comment 4.

This part of your response should be clearly stated in the methods speaking about which model was used

‘growth modelling in this paper was conducted exclusively to test whether the two types of

vertebrae provided the same results in terms of growth curves. For this reason, we decided to rely

only on the von Bertalanffy function, as it is more easily comparable with the existing literature.’

Its good to hear a robust age and growth study on the way. A point of caution is that the below part of the response is fundamentally incorrect; its all about fitting the best model to the length-at-age data at hand to produce the best fit for your data. Not exploring the best model for the data used in caution of setting a precedent does not make sense, and I don’t understand what is meant. Perhaps you are referring to the underlying sample size considering the follow up paper you mention. In anycase, the intention of a multimodel approach is not to one day revert back to a priori selection of a single model, just want to make sure that is clear.

“Moreover, considering the small sample size, we were concerned that applying a multi-model comparison could introduce a potential bias by indicating one model as “better” without a solid basis,”

This response has not addressed the use of t0 – a biologically meaningless parameter that can very easily be recalculated to L0 from Linf and K:

L0 = L∞(1− exp(k*t0))

Line 40

Would be good to add the age range here, as per previous review these results may differ for older animals “…were more precise for age classes … , offering..”

Line 117

This sentence is incomplete. State ‘draw conclusion on the most effective vertebrae preparation methods for the Raja genus’ or something similar

Line 240-44

I wonder if a measure of the number of published age and growth papers each reader has been an author on, an/or led or been senior author on, and/or been an ager in, and/or number of sp. aged would be better metrics? I for example in my first age and growth paper read about 1500 vertebrae of a single species in two populations before proceeding with age estimates for that paper (n>500). That means under the present definition I would have medium experience despite at that point, only having aged one species and not having had an accepted publication yet; I’d consider my experience low at that point. Something to consider. I understand the present system captures the spectrum of present experience, but it doesn’t confer much to actual robust contribution to the field as a measure of expertise.

Line 443

‘k’ in VBGF is not a growth rate but a coefficient of the curvature of the function that describes a proxy for the ‘growth completion rate’ determined from the curveature between L0 and Linf. Easiest to replace ‘growth rate’ with ‘k’ than to describe the nuances of this. It can be paired with statements like ‘higher k values indicates faster growth’. There are equations for calculating growth rate from VBGF parameters if wanting to pursue that, but I don’t see it as needed. It’s a common mistake in age and growth literature. In 4.1 discussion, and elsewhere please adjust use of 'growth rate' as a direct infernece from k, or substitute of using k.

**Do you want your identity to be public for this peer review?** For information about this choice, including consent withdrawal, please see our Privacy Policy

Reviewer #2: No

---

## [Author Response · Author response to Decision Letter 2]

13 Jan 2026

Rebuttal Letter of the manuscript PONE-D-25-37758 with the updated title ‘A comparison of vertebral preparation techniques for increasing reader precision and agreement in vertebral band pair identification of Northeast Atlantic skates’ submitted to the PLOS ONE.

Dear Joel Gayford,

We thank the reviewers for their final comments concerning the manuscript. We value their feedback and the manuscript has improved considerably from their input. We also updated our sample sizes where minor updates to the tables were also incorporated.

Reviewer 2 synopsis:

The reviewer recommends strengthening this justification further by explicitly embedding the rationale for using modal age within the Methods section, as it is a foundational methodological decision influencing interpretation of results. They also agree that restricting analyses to the von Bertalanffy growth function may be acceptable given the stated objectives, but stresses that this should also be clearly stated in the methods.

Reviewer #2: I thank the authors for their response to my comments and improvements made to the manuscript. At heart, I do not agree with the modal age approach, but I am satisfied that the first paragraph of the discussion outlines the key constraint for the reader, and the purpose of its use herein. There is scope to justify the reasoning for this approach in the methods additionally, as it is a foundational decision made in the conduct of the methodology of this study, and an action that influences the results and their interpretation (see below comment for a suggestion, as you’ve actually provided a really good couple of sentences in a reply to a comment).

1. Considering this text now added to the start of the discussion, and reply to my comment 3., I wonder whether the title is entirely appropriate. The final sentence of the first discussion paragraph for example is correct, and also implies the title is misleading

“Therefore, our finding that anterior, whole, unstained vertebrae provided the highest reader precision and agreement should be interpreted within this context”

A more appropriate title as such would be a synonym of:

A comparison of vertebral preparation techniques to increase reader precision and agreement in vertebral band pair identification in northwest Atlantic skates.

This is what the study has done. I recommend saving the ‘age estimation’ for the follow up manuscript that you mention

A few responses to previous comment reply’s and just a few more line comments below. Well done to the authors, this was quite a large review and I appreciate the time in constructing considered responses.

We agree with the reviewer and have updated the title as suggested.

2. Reply to comment 3.

The statement ‘modal age remains a well-established and widely accepted practice in age and growth validation studies’ is something I just do not consider to be true. Noting the authors list of referred to papers (Campana 2001; Goldman 2005; Vitale et al., 2021; ICES, 2022) in their response to comment 3., Campana 2001 for example speaks to ‘modal progression’ which is in the context of length class modes corresponding to ages, and as such, whether length modes in an annual breeding population can be used to estimate age from length; not taking the mode of age estimates across a range of readers who have examined banding patterns in calcified hard parts to determine age. I haven’t had much luck locating others cited, but ensure there is no confusion in ‘modal progression’ which is different to the present method. Regardless, better to justify the use of using modal ages to the readership more so than me. I like the justification provided in the first paragraph in the authors reply to the original comment 3., and would encourage its inclusion within the paragraph ending at line 194

‘…. modal age can potentially introduce a systematic underageing bias, particularly if the highest (oldest) age estimates more closely approximate true age. However, our primary objective was not to determine absolute accuracy but rather to assess how different preparation methods influence the readability and consistency of vertebral age estimates across multiple readers. The modal age approach enabled us to quantify reader consensus and exclude indeterminate cases (i.e., instances with no clear mode), thereby focusing the analysis on relative precision rather than absolute accuracy.’

We thank the reviewer for their comment and we have included the text in the manuscript as suggested.

3. Reply to comment 4.

This part of your response should be clearly stated in the methods speaking about which model was used

‘growth modelling in this paper was conducted exclusively to test whether the two types of

vertebrae provided the same results in terms of growth curves. For this reason, we decided to rely

only on the von Bertalanffy function, as it is more easily comparable with the existing literature.’

We agree with the reviewer and the sentence has also been included in the methods (now line 229-231).

4. Its good to hear a robust age and growth study on the way. A point of caution is that the below part of the response is fundamentally incorrect; its all about fitting the best model to the length-at-age data at hand to produce the best fit for your data. Not exploring the best model for the data used in caution of setting a precedent does not make sense, and I don’t understand what is meant. Perhaps you are referring to the underlying sample size considering the follow up paper you mention. In anycase, the intention of a multimodel approach is not to one day revert back to a priori selection of a single model, just want to make sure that is clear.

“Moreover, considering the small sample size, we were concerned that applying a multi-model comparison could introduce a potential bias by indicating one model as “better” without a solid basis,”

This response has not addressed the use of t0 – a biologically meaningless parameter that can very easily be recalculated to L0 from Linf and K:

L0 = L∞(1− exp(k*t0))

We understand the concern from the reviewer regarding t0 calculation. We have revised the Methods section to explicitly state that t₀ is retained solely as a mathematical curve-fitting parameter and is not interpreted biologically. The relevant text has been added to the growth modelling section:

“Curve fitting and growth model reliability near age 0, age–at–length data from eight R. clavata and ten R. montagui hatchlings born in captivity in the aquatic research facility of Wageningen University CARUS were added to the dataset. These observations were included used to constrain the early proportion of the growth curve, with the parameter t0 treated as a mathematical curve-fitting parameter.” (line 232-236)

5. Line 40

Would be good to add the age range here, as per previous review these results may differ for older animals “…were more precise for age classes … , offering..”

We agree with the reviewer and the sentence has been changed to “…were more precise for age classes 0 - 9 years, offering…”

6. Line 117

This sentence is incomplete. State ‘draw conclusion on the most effective vertebrae preparation methods for the Raja genus’ or something similar.

The sentence has been changed to “…we aim to draw conclusions on the most effective vertebral preparation techniques for the Raja genus”.

7. Line 240-44

I wonder if a measure of the number of published age and growth papers each reader has been an author on, an/or led or been senior author on, and/or been an ager in, and/or number of sp. aged would be better metrics? I for example in my first age and growth paper read about 1500 vertebrae of a single species in two populations before proceeding with age estimates for that paper (n>500). That means under the present definition I would have medium experience despite at that point, only having aged one species and not having had an accepted publication yet; I’d consider my experience low at that point. Something to consider. I understand the present system captures the spectrum of present experience, but it doesn’t confer much to actual robust contribution to the field as a measure of expertise.

We thank the Reviewer for raising an important and conceptually relevant point regarding the definition and quantification of experience. We acknowledge that the approach adopted has limitations and does not aim to fully capture all dimensions of expertise in age and growth studies. Nevertheless, we believe that, despite its imperfections, it represents a pragmatic and easily implementable solution that is not necessarily inappropriate for the objectives of the present study.

The use of alternative metrics, such as the number of published papers or the number of species included in previous studies, would not necessarily provide a more accurate representation of actual experience. For instance, an individual may be listed as an author on age and growth studies involving multiple species without having been directly involved in age estimation or in the interpretation of growth patterns, having instead contributed through sample provision or analyses conducted after age determination. In such cases, experience based on publication metrics would likely be overestimated.

Similarly, while experience gained across different species may ensure greater familiarity with the general ageing process, it may also introduce potential bias by unconsciously interpreting the growth patterns of a new species based on prior experience that is not necessarily transferable or appropriate in the new biological context. In light of these considerations, we consider the adopted criterion, while admittedly imperfect, to represent a reasonable compromise between simplicity, transparency, and internal consistency, and to avoid some of the potential distortions associated with more complex but not necessarily more informative metrics.

8. Line 443

‘k’ in VBGF is not a growth rate but a coefficient of the curvature of the function that describes a proxy for the ‘growth completion rate’ determined from the curveature between L0 and Linf. Easiest to replace ‘growth rate’ with ‘k’ than to describe the nuances of this. It can be paired with statements like ‘higher k values indicates faster growth’. There are equations for calculating growth rate from VBGF parameters if wanting to pursue that, but I don’t see it as needed. It’s a common mistake in age and growth literature. In 4.1 discussion, and elsewhere please adjust use of 'growth rate' as a direct infernece from k, or substitute of using k.

Instances of ‘growth rate’ were replaced with ‘growth coefficient’.

---

## [Editor Report · Decision Letter 2]

14 Jan 2026

A comparison of vertebral preparation techniques for increasing reader precision and agreement in vertebral band pair identification of Northeast Atlantic skates

PONE-D-25-37758R2

Dear Dr. Greenway,

We’re pleased to inform you that your manuscript has been judged scientifically suitable for publication and will be formally accepted for publication once it meets all outstanding technical requirements.

Kind regards,

Joel Harrison Gayford

Academic Editor

PLOS One
---

## [Editor Report · Acceptance letter]

PONE-D-25-37758R2

PLOS One

Dear Dr. Greenway,

I'm pleased to inform you that your manuscript has been deemed suitable for publication in PLOS One. Congratulations! Your manuscript is now being handed over to our production team.

Kind regards,

on behalf of

Mr. Joel Harrison Gayford

Academic Editor

PLOS One